# You Only Need Adversarial Supervision for Semantic Image Synthesis

**Edgar Schönfeld** *
Bosch Center for Artificial Intelligence

**Vadim Sushko** *
Bosch Center for Artificial Intelligence

**Dan Zhang**
Bosch Center for Artificial Intelligence

**Jürgen Gall**
University of Bonn

**Bernt Schiele**
Max Planck Institute for Informatics

**Anna Khoreva**
Bosch Center for Artificial Intelligence

## Abstract

Despite their recent successes, GAN models for semantic image synthesis still suffer from poor image quality when trained with only adversarial supervision. Historically, additionally employing the VGG-based perceptual loss has helped to overcome this issue, significantly improving the synthesis quality, but at the same time limiting the progress of GAN models for semantic image synthesis. In this work, we propose a novel, simplified GAN model, which needs only adversarial supervision to achieve high quality results. We re-design the discriminator as a semantic segmentation network, directly using the given semantic label maps as the ground truth for training. By providing stronger supervision to the discriminator as well as to the generator through spatially- and semantically-aware discriminator feedback, we are able to synthesize images of higher fidelity with better alignment to their input label maps, making the use of the perceptual loss superfluous. Moreover, we enable high-quality multi-modal image synthesis through global and local sampling of a 3D noise tensor injected into the generator, which allows complete or partial image change. We show that images synthesized by our model are more diverse and follow the color and texture distributions of real images more closely. We achieve an average improvement of 6 FID and 5 mIoU points over the state of the art across different datasets using only adversarial supervision.

Semantic label map    SPADE (Park et al., 2019) with VGG    w/o VGG    Our model (OASIS), sampled with different noise w/o VGG

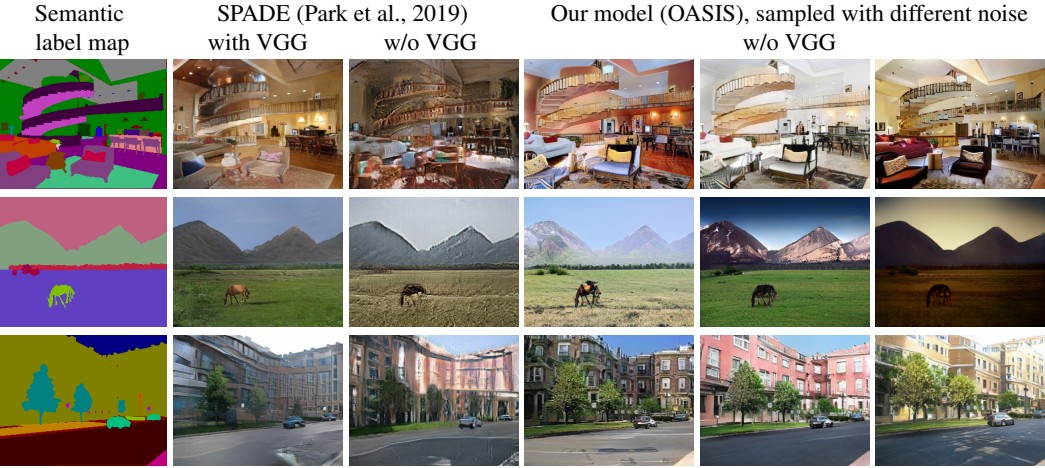

Figure 1: Existing semantic image synthesis models heavily rely on the VGG-based perceptual loss to improve the quality of generated images. In contrast, our model can synthesize diverse and high-quality images while only using an adversarial loss, without any external supervision.

---

*Equal contribution. Correspondence to {edgar.schoenfeld, vadim.sushko}@bosch.com.

## 1 INTRODUCTION

Conditional generative adversarial networks (GANs) (Mirza & Osindero, 2014) synthesize images conditioned on class labels (Zhang et al., 2019; Brock et al., 2019), text (Reed et al., 2016; Zhang et al., 2018a), other images (Isola et al., 2017; Huang et al., 2018), or semantic label maps (Wang et al., 2018; Park et al., 2019). In this work, we focus on the latter, addressing semantic image synthesis. Semantic image synthesis enables rendering of realistic images from user-specified layouts, without the use of an intricate graphic engine. Therefore, its applications range widely from content creation and image editing to generating training data that needs to adhere to specific semantic requirements (Wang et al., 2018; Chen & Koltun, 2017). Despite the recent progress on stabilizing GANs (Gulrajani et al., 2017; Miyato et al., 2018; Zhang & Khoreva, 2019) and developing their architectures (Zhang et al., 2019; Karras et al., 2019), state-of-the-art GAN-based semantic image synthesis models (Park et al., 2019; Liu et al., 2019) still greatly suffer from training instabilities and poor image quality when trained only with adversarial supervision (see Fig. 1). An established practice to overcome this issue is to employ a perceptual loss (Wang et al., 2018) to train the generator, in addition to the discriminator loss. The perceptual loss aims to match intermediate features of synthetic and real images, that are estimated via an external perception network. A popular choice for such a network is VGG (Simonyan & Zisserman, 2015), pre-trained on ImageNet (Deng et al., 2009). Although the perceptual loss substantially improves the accuracy of previous methods, it comes with the computational overhead introduced by utilizing an extra network for training. Moreover, it usually dominates over the adversarial loss during training, which can have a negative impact on the diversity and quality of generated images, as we show in our experiments. Therefore, in this work we propose a novel, simplified model that achieves state-of-the-art results without requiring a perceptual loss.

A fundamental question for GAN-based semantic image synthesis models is how to design the discriminator to efficiently utilize information from the given semantic label maps. Conventional methods (Park et al., 2019; Wang et al., 2018; Liu et al., 2019; Isola et al., 2017) adopt a multi-scale classification network, taking the label map as input along with the image, and making a global image-level real/fake decision. Such a discriminator has limited representation power, as it is not incentivized to learn high-fidelity pixel-level details of the images and their precise alignment with the input semantic label maps. To mitigate this issue, we propose an alternative architecture for the discriminator, re-designing it as an encoder-decoder semantic segmentation network (Ronneberger et al., 2015), and directly exploiting the given semantic label maps as ground truth via a ($N$+1)-class cross-entropy loss (see Fig. 3). This new discriminator provides semantically-aware pixel-level feedback to the generator, partitioning the image into segments belonging to one of the $N$ real semantic classes or the fake class. Enabled by the discriminator per-pixel response, we further introduce a LabelMix regularization, which fosters the discriminator to focus more on the semantic and structural differences of real and synthetic images. The proposed changes lead to a much stronger discriminator, that maintains a powerful semantic representation of objects, giving more meaningful feedback to the generator, and thus making the perceptual loss supervision superfluous (see Fig. 1).

Next, we propose to enable multi-modal synthesis of the generator via 3D noise sampling. Previously, directly using 1D noise as input was not successful for semantic image synthesis, as the generator tended to mostly ignore it or synthesized images of poor quality (Isola et al., 2017; Wang et al., 2018). Thus, prior work (Wang et al., 2018; Park et al., 2019) resorted to using an image encoder to produce multi-modal outputs. In this work, we propose a lighter solution. Empowered by our stronger discriminator, the generator can effectively synthesize different images by simply re-sampling a 3D noise tensor, which is used not only as the input but also combined with intermediate features via conditional normalization at every layer. Such noise is spatially sensitive, so we can re-sample it both globally (channel-wise) and locally (pixel-wise), allowing to change not only the appearance of the whole scene, but also of specific semantic classes or any chosen areas (see Fig. 2). We call our model OASIS, as it needs **o**nly **a**dversarial **s**upervision for **s**emantic **i**mage **s**ynthesis.

In summary, our main contributions are: (1) We propose a novel segmentation-based discriminator architecture, that gives more powerful feedback to the generator and eliminates the necessity of the perceptual loss supervision. (2) We present a simple 3D noise sampling scheme, notably increasing the diversity of multi-modal synthesis and enabling complete or partial change of the generated image. (3) With the OASIS model, we achieve high quality results on the ADE20K, Cityscapes and COCO-stuff datasets, on average improving the state of the art by 6 FID and 5 mIoU points, while

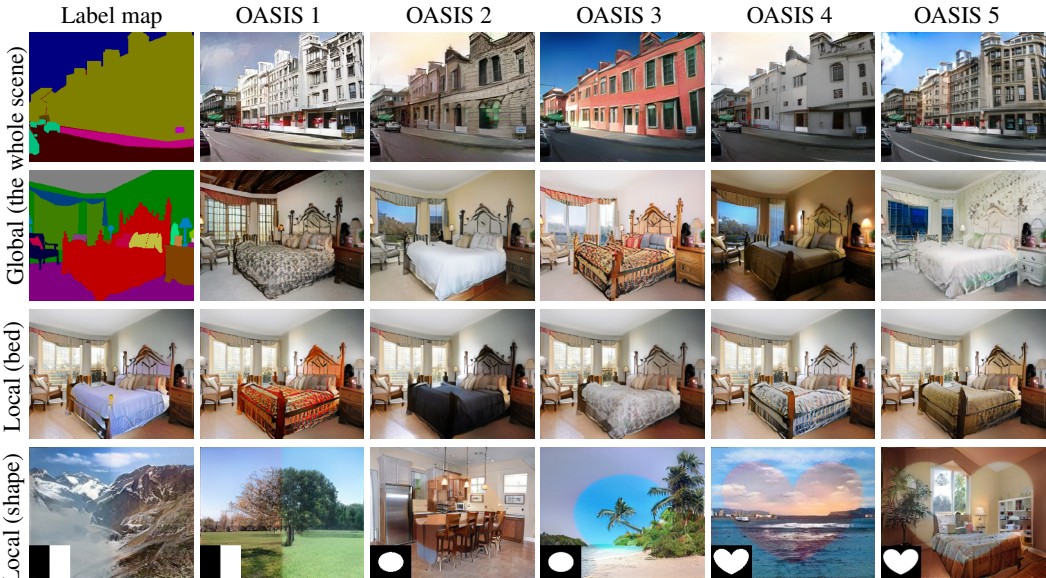

Figure 2: OASIS multi-modal synthesis results. The 3D noise can be sampled globally (first 2 rows), changing the whole scene, or locally (last 2 rows), partially changing the image. For the latter, we sample different noise per region, like the bed segment (in red) or arbitrary areas defined by shapes.

relying only on adversarial supervision. We show that images synthesized by OASIS exhibit much higher diversity and more closely follow the color and texture distributions of real images. Our code and pretrained models are available at `https://github.com/boschresearch/OASIS`.

## 2  RELATED WORK

**Semantic image synthesis.** Pix2pix (Isola et al., 2017) first proposed to use conditional GANs (Mirza & Osindero, 2014) for semantic image synthesis, adopting an encoder-decoder generator which takes semantic label maps as input, and employing a PatchGAN discriminator. Since then, various generator and discriminator modifications have been introduced (Wang et al., 2018; Park et al., 2019; Liu et al., 2019; Tang et al., 2020c;b; Ntavelis et al., 2020). Besides GANs, Chen & Koltun (2017) proposed to use a cascaded refinement network (CRN) for high-resolution semantic image synthesis, and SIMS (Qi et al., 2018) extended it with a non-parametric component, serving as a memory bank of source material to assist the synthesis. Further, Li et al. (2019) employed implicit maximum likelihood estimation (Li & Malik, 2018) to increase the variety of the CRN model. However, these approaches still underperform in comparison to state-of-the-art GAN models. Therefore, next we focus on the recent GAN architectures for semantic image synthesis.

**Discriminator architectures.** Pix2pix (Isola et al., 2017), Pix2pixHD (Wang et al., 2018) and SPADE (Park et al., 2019) all employed a multi-scale PatchGAN discriminator, that takes an image and its semantic label map as input. CC-FPSE (Liu et al., 2019) proposed a feature-pyramid discriminator, embedding both images and label maps into a joint feature map, and then consecutively upsampling it in order to classify it as real/fake at multiple scales. LGGAN (Tang et al., 2020c) introduced a classification-based feature learning module to learn more discriminative and class-specific features. In this work, we propose to use a pixel-wise semantic segmentation network as a discriminator instead of multi-scale image classifiers as in the above approaches, and to directly exploit the semantic label maps for its supervision. Segmentation-based discriminators have been shown to improve semantic segmentation (Souly et al., 2017) and unconditional image synthesis (Schönfeld et al., 2020), but to the best of our knowledge have not been explored for semantic image synthesis and our work is the first to apply adversarial semantic segmentation loss for this task.

**Generator architectures.** Conventionally, the semantic label map is provided to the image generation pipeline via an encoder (Isola et al., 2017; Wang et al., 2018; Tang et al., 2020c;b; Ntavelis et al., 2020). However, it is shown to be suboptimal at preserving the semantic information until the later stages of image generation. Therefore, SPADE introduced a spatially-adaptive normalization layer

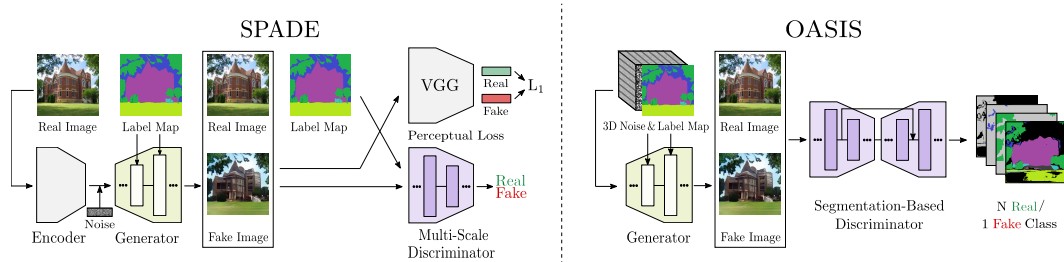

Figure 3: SPADE (left) vs. OASIS (right). OASIS outperforms SPADE, while being simpler and lighter: it uses only adversarial loss supervision and a single segmentation-based discriminator, without relying on heavy external networks. Furthermore, OASIS learns to synthesize multi-modal outputs by directly re-sampling the 3D noise tensor, instead of using an image encoder as in SPADE.

that directly modulates the label map onto the generator's hidden layer outputs at various scales. Alternatively, CC-FPSE proposed to use spatially-varying convolution kernels conditioned on the label map. Struggling with generating diverse images from noise, both Pix2pixHD and SPADE resorted to having an image encoder in the generator design to enable multi-modal synthesis. The generator then combines the extracted image style with the label map to reconstruct the original image. By alternating the style vector, one can generate multiple outputs conditioned on the same label map. However, using an image encoder is a resource demanding solution. In this work, we enable multi-modal synthesis directly through sampling of a 3D noise tensor injected at every layer of the network. Differently from structured noise injection of Alharbi & Wonka (2020) and class-specific latent codes of Zhu et al. (2020), we inject the 3D noise along with label maps and adjust it to image resolution, also enabling re-sampling of selected semantic segments (see Fig. 2).

**Perceptual losses.** Gatys et al. (2015); Gatys et al. (2016); Johnson et al. (2016) and Bruna et al. (2016) were pioneers at exploiting perceptual losses to produce high-quality images for super-resolution and style transfer using convolutional networks. For semantic image synthesis, the VGG-based perceptual loss was first introduced by CRN, and later adopted by Pix2pixHD. Since then, it has become a default for training the generator (Park et al., 2019; Liu et al., 2019; Tan et al., 2020; Tang et al., 2020a). As the perceptual loss is based on a VGG network pre-trained on ImageNet (Deng et al., 2009), methods relying on it are constrained by the ImageNet domain and the representational power of VGG. With the recent progress on GAN training, e.g. by architecture designs and regularization techniques, the actual necessity of the perceptual loss requires a reassessment. We experimentally show that such loss imposes unnecessary constraints on the generator, significantly limiting sample diversity. While our model, trained without the VGG loss, achieves improved image diversity while not compromising image quality.

## 3 OASIS MODEL

In this section, we present our OASIS model, which, in contrast to other semantic image synthesis methods, needs only adversarial supervision for generator training. Using SPADE as a starting point (Sec. 3.1), we first propose to re-design the discriminator as a semantic segmentation network, directly using the given semantic label maps as ground truth (Sec. 3.2). Empowered by spatially- and semantically-aware feedback of the new discriminator, we next re-design the SPADE generator, enabling its effective multi-modal synthesis via 3D noise sampling (Sec. 3.3).

### 3.1 THE SPADE BASELINE

We choose SPADE as our baseline as it is a state-of-the-art model and a relatively simple representative of conventional semantic image synthesis models. As depicted in Fig. 3, the discriminator of SPADE largely follows the PatchGAN multi-scale discriminator (Isola et al., 2017), adopting two image classification networks operating at different resolutions. Both of them take the channel-wise concatenation of the semantic label map and the real/synthesized image as input, and produce true/-fake classification scores. On the generator side, SPADE adopts spatially-adaptive normalization layers to effectively integrate the semantic label map into the synthesis process from low to high scales. Additionally, the image encoder is used to extract the style vector from the reference image and then combine it with a 1D noise vector for multi-modal synthesis. The training loss of SPADE

consists of three terms, namely, an adversarial loss, a feature matching loss and the VGG-based perceptual loss: $\mathcal{L} = \max_G \min_D \mathcal{L}_{\text{adv}} + \lambda_{\text{fm}}\mathcal{L}_{\text{fm}} + \lambda_{\text{vgg}}\mathcal{L}_{\text{vgg}}$. Overall, SPADE is a resource demanding model at both training and test time, i.e., with two PatchGAN discriminators, an image encoder in addition to the generator, and the VGG loss. In the following, we revisit its architecture and introduce a simpler and more efficient model that offers better performance with less complexity.

## 3.2 OASIS DISCRIMINATOR

For the generator to learn to synthesize images that are well aligned with the input semantic label maps, we need a powerful discriminator that coherently captures discriminative semantic features at different image scales. While classification-based discriminators, such as PatchGAN, take label maps as input concatenated to images, they can afford to ignore them and make the decision solely on image patch realism. Thus, we propose to cast the discriminator task as a multi-class semantic segmentation problem to directly utilize label maps for supervision, and accordingly alter its architecture to an encoder-decoder segmentation network (see Fig. 3). Encoder-decoder networks have proven to be effective for semantic segmentation (Badrinarayanan et al., 2016; Chen et al., 2018). Thus, we build our discriminator architecture upon U-Net (Ronneberger et al., 2015), which consists of the encoder and decoder connected by skip connections. This discriminator architecture is multi-scale through its design, integrating information over up- and down-sampling pathways and through the encoder-decoder skip connections. For details on the architecture see App. C.1.

The segmentation task of the discriminator is formulated to predict the per-pixel class label of the real images, using the given semantic label maps as ground truth. In addition to the $N$ semantic classes from the label maps, all pixels of the fake images are categorized as one extra class. Overall, we have $N+1$ classes in the semantic segmentation problem, and thus propose to use a $(N+1)$-class cross-entropy loss for training. Considering that the $N$ semantic classes are usually imbalanced and that the per-pixel size of objects varies for different semantic classes, we weight each class by its inverse per-pixel frequency, giving rare semantic classes more weight. In doing so, the contributions of each semantic class are equally balanced, and, thus, the generator is also encouraged to adequately synthesize less-represented classes. Mathematically, the new discriminator loss is expressed as:

$$\mathcal{L}_D = -\mathbb{E}_{(x,t)}\left[\sum_{c=1}^{N}\alpha_c \sum_{i,j}^{H\times W} t_{i,j,c}\log D(x)_{i,j,c}\right] - \mathbb{E}_{(z,t)}\left[\sum_{i,j}^{H\times W}\log D(G(z,t))_{i,j,c=N+1}\right], \quad (1)$$

where $x$ denotes the real image; $(z,t)$ is the noise-label map pair used by the generator $G$ to synthesize a fake image; and the discriminator $D$ maps the real or fake image into a per-pixel $(N+1)$-class prediction probability. The ground truth label map $t$ has three dimensions, where the first two correspond to the spatial position $(i,j) \in H \times W$, and the third one is a one-hot vector encoding the class $c \in \{1,..,N+1\}$. The class balancing weight $\alpha_c$ is the inverse of the per-pixel class frequency

$$\alpha_c = \frac{H \times W}{\sum_{i,j}^{H\times W} E_t\left[\mathbb{1}[t_{i,j,c} = 1]\right]}. \quad (2)$$

**LabelMix regularization.** In order to encourage our discriminator to focus on differences in content and structure between the fake and the real classes, we propose a LabelMix regularization. Based on the semantic layout, we generate a binary mask $M$ to mix a pair $(x, \hat{x})$ of real and fake images conditioned on the same label map: $\text{LabelMix}(x, \hat{x}, M) = M \odot x + (1 - M) \odot \hat{x}$, as visualized in Fig. 4. Given the mixed image, we further train the discriminator to be equivariant under the LabelMix operation. This is achieved by adding a consistency loss term $\mathcal{L}_{cons}$ to Eq. 1:

$$\mathcal{L}_{cons} = \left\|D_{\text{logits}}\Big(\text{LabelMix}(x, \hat{x}, M)\Big) - \text{LabelMix}\Big(D_{\text{logits}}(x), D_{\text{logits}}(\hat{x}), M\Big)\right\|^2, \quad (3)$$

where $D_{\text{logits}}$ are the logits attained before the last softmax activation layer, and $\|\cdot\|$ is the $L_2$ norm. This consistency loss compares the output of the discriminator on the LabelMix image with the LabelMix of its outputs, penalizing the discriminator for inconsistent predictions. LabelMix is different to CutMix (Yun et al., 2019), which randomly samples the binary mask $M$. A random mask will introduce inconsistency between the pixel-level classes and the scene layout provided by the label map. For an object with the semantic class $c$, it will contain pixels from both real and fake images, resulting in two labels, i.e. $c$ and $N + 1$. To avoid such inconsistency, the mask of LabelMix is generated according to the label map, providing natural borders between semantic regions, see Fig. 4 (Mask $M$). Under LabelMix regularization, the generator is encouraged to respect the natural semantic boundaries, improving pixel-level realism while also considering the class segment shapes.

Label map     Real image $x$     Fake image $\hat{x}$     Mask $M$     LabelMix$_{(x,\hat{x})}$   $D_{\text{LabelMix}_{(x,\hat{x})}}$   LabelMix$_{(D_x,D_{\hat{x}})}$

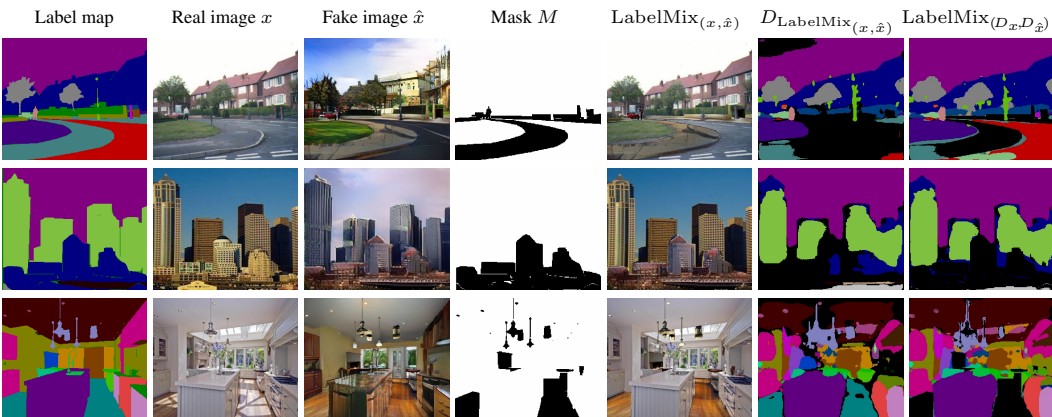

Figure 4: LabelMix regularization. Real $x$ and fake $\hat{x}$ images are mixed using a binary mask $M$, sampled based on the label map, resulting in LabelMix$_{(x,\hat{x})}$. The consistency regularization then minimizes the L2 distance between the logits of $D_{\text{LabelMix}_{(x,\hat{x})}}$ and LabelMix$_{(D_x,D_{\hat{x}})}$. In this visualization, **black** corresponds to the fake class in the $N+1$ segmentation output.

**Other variants.** Besides the proposed ($N$+1)-class cross entropy loss, there are other ways to train the segmentation-based discriminator with the label map. One can concatenate the label map to the input image, analogous to SPADE. Another option is to use projection, by taking the inner product between the last linear layer output and the embedded label map, analogous to class-label conditional GANs (Miyato & Koyama, 2018). For both alternatives, the training loss is pixel-level real/fake binary cross-entropy (Schönfeld et al., 2020). From the label map encoding perspective, these two variants use labels map as input (concatenated to image or at last linear layer), propagating it *forward* through the network. The ($N$+1)-setting uses the label map for loss computation, so it is propagated *backward* via gradient updates. Backward propagation ensures that the discriminator learns semantic-aware features, in contrast to forward propagation, where the label map alignment is not as strongly enforced. Performance comparison of the label map encodings is shown in Table 5.

### 3.3 OASIS GENERATOR

To stay in line with the OASIS discriminator design, the training loss for the generator is changed to

$$\mathcal{L}_G = -\mathbb{E}_{(z,t)}\left[\sum_{c=1}^{N}\alpha_c\sum_{i,j}^{H \times W}t_{i,j,c}\log D(G(z,t))_{i,j,c}\right], \tag{4}$$

which is a direct outcome of the non-saturation trick (Goodfellow et al., 2014) to Eq. 1. We next re-design the generator to enable multi-modal synthesis through noise sampling. SPADE is deterministic in its default setup, but can be trained with an extra image encoder to generate multi-modal outputs. We introduce a simpler version, that enables synthesis of diverse outputs directly from input noise. For this, we construct a noise tensor of size $64 \times H \times W$, matching the spatial dimensions of the label map $H \times W$. Channel-wise concatenation of the noise and label map forms a 3D tensor used as input to the generator and also as a conditioning at every spatially-adaptive normalization layer. In doing so, intermediate feature maps are conditioned on both the semantic labels and the noise (see Fig. 3). With such a design, the generator produces diverse, noise-dependent images. As the 3D noise is channel- and pixel-wise sensitive, at test time, one can sample the noise globally, per-channel, and locally, per-segment or per-pixel, for controlled synthesis of the whole scene or of specific semantic objects. For example, when generating a scene of a bedroom, one can re-sample the noise locally and change the appearance of the bed alone (see Fig. 2). Note that for simplicity during training we sample the 3D noise tensor globally, i.e. per-channel, replicating each channel value spatially along the height and width of the tensor. We analyse alternative ways of sampling 3D noise during training in App. A.7. Using image styles via an encoder, as in SPADE, is also possible in our setting, by replacing noise with encoder features. Lastly, to further reduce the complexity, we remove the first residual block in the generator, reducing the number of parameters from 96M to 72M (see App. C.2) without a noticeable performance loss (see Table 3).

| Label map | Ground truth | Pix2pixHD | SPADE | CC-FPSE | OASIS |

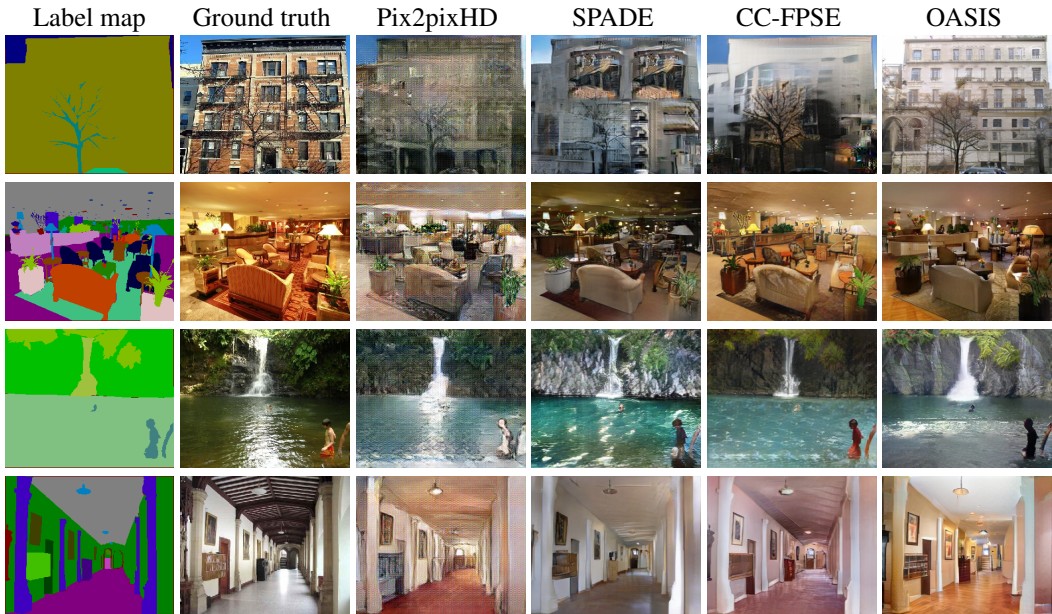

Figure 5: Qualitative comparison of OASIS with other methods on ADE20K. Trained with only adversarial supervision, our model generates images with better perceptual quality and structure.

## 4 EXPERIMENTS

We conduct experiments on three challenging datasets: ADE20K (Zhou et al., 2017), COCO-stuff (Caesar et al., 2018) and Cityscapes (Cordts et al., 2016). Following Qi et al. (2018), we also evaluate OASIS on ADE20K-outdoors, a subset of ADE20K containing outdoor scenes. We follow the experimental setting of Park et al. (2019). We did not use the GAN feature matching loss for OASIS, as we did not observe any improvement with it (see App. A.5), and used the VGG loss only for ablations with $\lambda_{\text{VGG}} = 10$. We did not experience any training instabilities and, thus, did not employ any extra stabilization techniques. All our models use an exponential moving average (EMA) of the generator weights with 0.9999 decay. For further training details refer to App. C.3.

Following prior work (Isola et al., 2017; Wang et al., 2018; Park et al., 2019; Liu et al., 2019), we evaluate models quantitatively on the validation set using the Fréchet Inception Distance (FID) (Heusel et al., 2017) and mean Intersection-over-Union (mIoU). FID is known to be sensitive to both quality and diversity and has been shown to be well aligned with human judgement (Heusel et al., 2017). We show additional evaluation of quality and diversity with "improved precision and recall" in App. A.9. Mean IoU is used to assess the alignment of the generated image with the ground truth label map, computed via a pre-trained semantic segmentation network. We use Uper-Net101 (Xiao et al., 2018) for ADE20K, multi-scale DRN-D-105 (Yu et al., 2017) for Cityscapes, and DeepLabV2 (Chen et al., 2015) for COCO-Stuff. We additionally propose to compare color and texture statistics between generated and real images on ADE20K to better understand how the perceptual loss influences performance. For this, we compute color histograms in LAB space and measure the earth mover's distance between the real and generated sets (Rubner et al., 2000). We measure the texture similarity to the real data as the $\chi^2$-distance between Local Binary Patterns histograms (Ojala et al., 1996). As different classes have different color and texture distributions, we aggregate histogram distances separately per class and then take the mean. Lower values for the texture and color distances indicate a closer similarity to real data.

### 4.1 MAIN RESULTS

We use SPADE as our baseline, using the authors' implementation[1]. For a fair comparison, we train this model without the feature matching loss and using EMA (Yaz et al., 2018) at test phase, which

---

[1]github.com/NVlabs/SPADE

Table 1: Comparison with other methods across datasets. Bold denotes the best performance.

| Method | # param | VGG | ADE20K FID↓ | ADE20K mIoU↑ | ADE-outd. FID↓ | ADE-outd. mIoU↑ | Cityscapes FID↓ | Cityscapes mIoU↑ | COCO-stuff FID↓ | COCO-stuff mIoU↑ |
|---|---|---|---|---|---|---|---|---|---|---|
| CRN | 84M | ✓ | 73.3 | 22.4 | 99.0 | 16.5 | 104.7 | 52.4 | 70.4 | 23.7 |
| SIMS | 56M | ✓ | n/a | n/a | 67.7 | 13.1 | 49.7 | 47.2 | n/a | n/a |
| Pix2pixHD | 183M | ✓ | 81.8 | 20.3 | 97.8 | 17.4 | 95.0 | 58.3 | 111.5 | 14.6 |
| LGGAN | n/a | ✓ | 31.6 | 41.6 | n/a | n/a | 57.7 | 68.4 | n/a | n/a |
| CC-FPSE | 131M | ✓ | 31.7 | 43.7 | n/a | n/a | 54.3 | 65.5 | 19.2 | 41.6 |
| SPADE | 102M | ✓ | 33.9 | 38.5 | 63.3 | 30.8 | 71.8 | 62.3 | 22.6 | 37.4 |
| SPADE+ | 102M | ✓ | 32.9 | 42.5 | 51.1 | 32.1 | 47.8 | 64.0 | 21.7 | 38.8 |
| | | ✗ | 60.7 | 21.0 | 65.4 | 22.7 | 61.4 | 47.6 | 99.1 | 16.1 |
| OASIS | 94M | ✗ | **28.3** | **48.8** | **48.6** | **40.4** | **47.7** | **69.3** | **17.0** | **44.1** |

Table 2: Multi-modal synthesis evaluation on ADE20K. Bold and red denote the best and the worst performance, respectively.

| Method | Multi-mod. | VGG | MS-SSIM↓ | LPIPS↑ | FID↓ | mIoU↑ |
|---|---|---|---|---|---|---|
| SPADE+ | Encoder | ✓ | 0.85 | 0.16 | 33.4 | 40.2 |
| SPADE+ | 3D noise | ✗ | **0.35** | **0.50** | 58.4 | 18.7 |
| | | ✓ | 0.53 | 0.36 | 34.4 | 36.2 |
| OASIS | 3D noise | ✗ | 0.65 | 0.35 | **28.3** | 48.8 |
| | | ✓ | 0.88 | 0.15 | 31.6 | **50.8** |

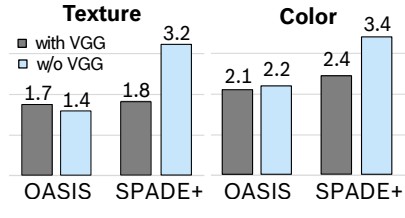

Figure 6: Histogram distances to real data.

we further refer to as SPADE+. We found that the feature matching loss has a negligible impact (see App. A.5), while EMA significantly increases the performance for all metrics (see Table 1).

OASIS outperforms the current state of the art on all datasets with an average improvement of 6 FID and 5 mIoU points (Table 1). Importantly, OASIS achieved the improvement via adversarial supervision alone. On the contrary, SPADE+ does not produce images of high visual quality without the perceptual loss, and struggles to learn the color and texture distribution of real images (Fig. 6). A strong discriminator is the key factor for good performance: without a rich training signal from the discriminator, the SPADE+ generator has to learn through minimizing the VGG loss. With the stronger OASIS discriminator, the perceptual loss does not overtake the generator supervision (see App. A.2), allowing to produce images with the color and texture distribution closer to the real data.

Fig. 5 shows a qualitative comparison of our results to previous models. Our approach noticeably improves image quality, synthesizing finer textures and more natural colors. With the powerful feedback from the discriminator, OASIS is able to learn the appearance of small or rarely occurring semantic classes (which is reflected in the per-class IoU scores presented in App. A.3), producing plausible results even for complex scenes with rare classes and reducing unnatural artifacts.

**Multi-modal image synthesis.** In contrast to previous work, OASIS can produce diverse images by directly re-sampling input 3D noise. As 3D noise modulates features directly at every layer of the generator at different scales, matching their resolution, it affects both global and local characteristics of the image. Thus, the noise can be sampled globally, varying the whole image, or locally, resulting in the selected object change while preserving the rest of the scene (see Fig. 2).

To measure the variation in the multi-modal generation, we evaluate MS-SSIM (Wang et al., 2003) and LPIPS (Zhang et al., 2018b) between images generated from the same label map. We generate 20 images and compute the mean pairwise scores, and then average over all label maps. The lower the MS-SSIM and the higher the LPIPS scores, the more diverse the generated images are. To assess the effect of the perceptual loss and the noise sampling on diversity, we train SPADE+ with 3D noise or the image encoder, and with or without the perceptual loss. Table 2 shows that OASIS, without perceptual loss, improves over SPADE+ with the image encoder, both in terms of image diversity (MS-SSIM, LPIPS) and quality (mean FID, mIoU across 20 realizations). Using 3D noise further increases diversity for SPADE+. However, a strong quality-diversity tradeoff exists for SPADE+: 3D noise improves diversity at the cost of quality, and the perceptual loss improves quality at the cost of diversity. For OASIS, the VGG loss also reduces diversity but does not noticeably affect quality. Note that in our experiments LabelMix does not notably affect diversity (see App. A.1).

## 4.2 ABLATIONS

We conduct ablations on ADE20K to evaluate our proposed changes. The main ablation shows the impact of our new discriminator, lighter generator, LabelMix and 3D noise. Further ablations are concerned with architecture changes and the label map encodings in the discriminator, where for fair comparison we use no 3D noise and LabelMix.

**Main ablation.** Table 3 shows that SPADE+ scores low on the image quality metrics without the perceptual loss. Replacing the SPADE+ discriminator with the OASIS discriminator, while keeping the generator fixed, improves FID and mIoU by more than 30 points. Changing the SPADE+ generator to the lighter OASIS generator leads to a negligible degradation of $0.3$ in FID and $0.5$ in mIoU. With LabelMix FID improves further by $\sim 1$ point (more ablations on LabelMix in App. A.4). Adding 3D noise

Table 3: OASIS ablation on ADE20K. Bold denotes the best performance.

| $G$ | $D$ | VGG | LabelMix | FID↓ | mIoU↑ |
|---|---|---|---|---|---|
| SPADE+ | SPADE+ | ✗ | ✗ | 60.7 | 21.0 |
| SPADE+ | OASIS | ✗ | ✗ | 29.0 | **52.1** |
| OASIS | OASIS | ✗ | ✗ | 29.3 | 51.6 |
| | | ✗ | ✓ | 28.4 | 50.6 |
| OASIS | OASIS | ✗ | ✓ | **28.3** | 48.8 |
| +3D noise | | ✓ | ✓ | 31.6 | 50.8 |

improves FID but degrades mIoU, as diversity complicates the task of the pre-trained semantic segmentation network used to compute the score. For OASIS the perceptual loss deteriorates FID by more than 2 points, but improves mIoU. Overall, without the perceptual loss the new discriminator is the key to the performance boost over SPADE+.

**Ablation on the discriminator architecture.** We train the OASIS generator with three alternative discriminators: the original multi-scale PatchGAN consisting of two networks, a single-scale PatchGAN, and a ResNet-based discriminator, corresponding to the encoder of the U-Net shaped OASIS discriminator. Table 4 shows that the alternative discriminators only perform well with perceptual supervision, while the OASIS discriminator achieves superior performance independent of it. The single-scale

Table 4: Ablation on the $D$ architecture. Bold denotes the best performance, red highlights collapsed runs.

| $D$ architecture | w/o VGG | | with VGG | |
|---|---|---|---|---|
| | FID↓ | mIoU↑ | FID↓ | mIoU↑ |
| MS-PatchGAN (2x) | 60.7 | 21.0 | 32.9 | 42.5 |
| PatchGAN | 197 | 0.62 | 34.2 | 42.2 |
| ResNet-PatchGAN | 147 | 0.42 | 32.4 | 45.1 |
| OASIS | **29.3** | **51.6** | **29.2** | **51.1** |

discriminators even collapse without the perceptual loss (highlighted in red in Table 4).

**Ablation on the label map encoding.** We study four different label map encodings: input concatenation, as in SPADE, projection conditioned on the label map (Miyato & Koyama, 2018), employing label maps as ground truth for the $N+1$ segmentation loss, or for the class-balanced $N+1$ loss (see Sec. 3.2). As shown in Table 5, input concatenation is not sufficient without additional perceptual loss supervision, leading to training collapse. Without perceptual loss, the $N+1$ loss outperforms the input con-

Table 5: Ablation on the label map encoding. Bold denotes the best performance, red highlights collapsed runs.

| Label encoding | w/o VGG | | with VGG | |
|---|---|---|---|---|
| | FID↓ | mIoU↑ | FID↓ | mIoU↑ |
| Input concatenation | 280 | 0.02 | 30.0 | 43.9 |
| Projection | 32.4 | 44.9 | **28.0** | 46.9 |
| N+1 loss | **28.3** | 47.2 | 28.6 | 49.8 |
| Balanced N+1 loss | 29.3 | **51.6** | 29.2 | **51.1** |

catenation and the projection in both the FID and mIoU metrics. The class balancing noticeably improves mIoU due to better supervision for rarely occurring semantic classes. More ablations can be found in App. A.

## 5 CONCLUSION

In this work we propose OASIS, a semantic image synthesis model that only relies on adversarial supervision to achieve high fidelity image synthesis. In contrast to previous work, our model eliminates the need for a perceptual loss, which often imposes extra constraints on image quality and diversity. This is achieved via detailed spatial and semantic-aware supervision from our novel segmentation-based discriminator, which uses semantic label maps as ground truth for training. With this powerful discriminator, OASIS can easily generate diverse multi-modal outputs by re-sampling the 3D noise, both globally and locally, allowing to change the appearance of the whole scene and of individual objects. OASIS significantly improves over the state of the art in terms of image quality and diversity, while being simpler and more lightweight than previous methods.

## ACKNOWLEDGEMENT

Jürgen Gall has been supported by the Deutsche Forschungsgemeinschaft (DFG, German Research Foundation) under Germany's Excellence Strategy - EXC 2070 -390732324.

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

APPENDIX

This supplementary material to the main paper is structured as follows:

## A  QUANTITATIVE RESULTS

### A.1  SUMMARIZED MAIN ABLATION OVER TWO DATASETS

Table A: Summarized ablation on two datasets. Bold denotes the best performance. Red denotes the worst performance among experiments with 3D noise. Green denotes the major performance gains that are caused by the proposed OASIS discriminator and LabelMix.

| Method | Cityscapes | | | ADE20K | | |
|---|---|---|---|---|---|---|
| | FID↓ | mIoU↑ | MS-SSIM↓ | FID↓ | mIoU↑ | MS-SSIM↓ |
| SPADE+ | 61.4 | 47.6 | 1.0 | 60.7 | 21.0 | 1.0 |
| + OASIS D, G | 54.1 | 67.6 | 1.0 | 29.3 | 51.6 | 1.0 |
| + 3D noise | 51.5 | 66.3 | **0.62** | 28.9 | 47.3 | **0.63** |
| + LabelMix | 47.7 | 69.3 | 0.64 | **28.3** | 48.8 | 0.65 |
| + VGG | **46.1** | **72.0** | *0.84* | 31.6 | 50.8 | *0.88* |

In Table A we present a summarized version of our ablations for the ADE20K and Cityscapes dataset. The following observations can be made:

(1) Looking at the 2nd row of Table A, we see that the main performance gain comes from the discriminator design (major) (OASIS D,G). The OASIS generator is a lighter version of the SPADE generator, which does not result in a performance improvement (Table 3), but has significantly less parameters. A second source of improvement is LabelMix.

(2) The mIoU can drop when 3D noise is added, as diversity complicates the task of the pre-trained semantic segmentation network that is used to compute the mIoU score. Note that the purpose of noise is not to improve the image quality (FID) but to improve diversity (MS-SSIM).

(3) The perceptual loss can hurt performance and diversity by biasing the generator towards ImageNet, as in this case the target distribution is more difficult to recreate fully. By punishing diversity, the perceptual loss encourages generating images with more standard semantic features This facilitates the task of external pretrained segmenters, and consequently helps to raise the mIoU metric.

## A.2    THE INFLUENCE OF VGG ON TRAINING DYNAMICS

Table 1 and Figure 1 illustrate that performance of SPADE+ strongly depends on the perceptual loss. In contrast, OASIS achieves high quality without this loss (Table 1). We find the explanation in the fact, that the SPADE+ Patch-GAN discriminator does not provide a strong training signal for the generator. At the absence of strong supervision from the discriminator, the generator resorts to learning mostly from the VGG loss. The loss curves in Fig. A support this finding: throughout the training the SPADE+ model focuses on minimizing the VGG loss, keeping the adversarial generator loss more or less constant. In contrast, OASIS significantly improves adversarial generator loss during training, learning to fool the segmentation-based OASIS discriminator. That indicates a better adversarial balance, when the generator learns semantically meaningful features that the segmenter judges as real. The difference in scales of G loss for models comes from different objectives, since SPADE+ optimizes binary cross entropy, and OASIS minimizes multi-class cross entropy with $N+1$ classes.

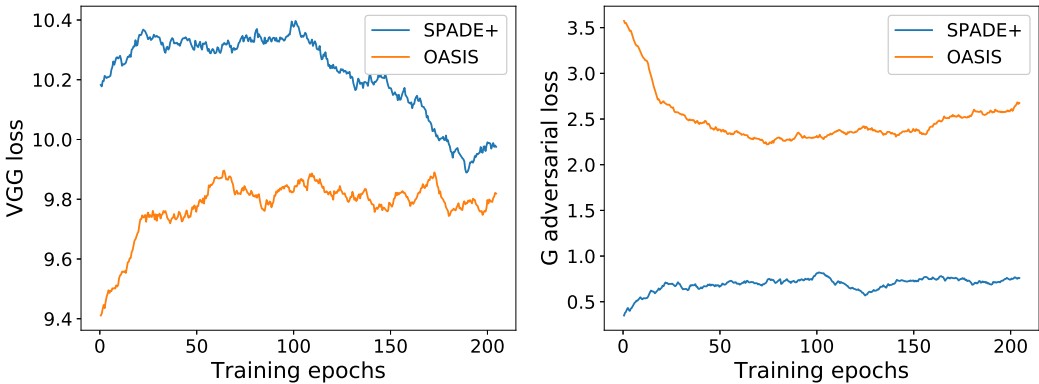

Figure A:  VGG and adversarial G losses for SPADE and OASIS, trained with the perceptual loss

## A.3    PER-CLASS IOU SCORES

As seen in Table 1 in the main paper, OASIS significantly outperforms previous approaches in mIoU. We found that the improvement comes mainly from the better IoU scores achieved for less-represented semantic classes. To illustrate the gain, we report per-class IoU scores on ADE20k, COCO-Stuff and Cityscapes in Tables B, C and D. For visualization purposes, we sorted the semantic classes of all datasets, ordering by their pixel-wise frequency in the training images.

Taking ADE20k as an example, Table B highlights that the relative gain in mIoU is especially high for the group of less-represented semantic classes, that cover less than 3% of all the images. For these rare classes the relative gain over the baseline exceeds 40%. We found that the gain majorly comes from the per-class balancing applied in the OASIS loss function. In order to illustrate this effect, we train OASIS without the proposed balancing. Table B reveals, this baseline reaches a bit higher score for frequent classes, but shows worse performance for the rarely occurring ones. This is expected, as the balancing down-weights the objects met frequently while up-weights infrequent classes. We thus conclude that the balancing draws the attention of the discriminator to rarely occurring semantic classes, which results in a much higher quality of the generation.

## A.4    ABLATION ON LABELMIX

Consistency regularization for the segmentation output of the discriminator requires a method of generating binary masks. Therefore, we compare the effectiveness of CutMix (Yun et al., 2019) and our proposed LabelMix. Both methods produce binary masks, but only LabelMix respects the boundaries between semantic classes in the label map. Table E compares the FID and mIoU scores of OASIS trained with both methods on the Cityscapes dataset. It can be seen that LabelMix improves both FID (51.5 vs. 47.7) and mIoU (66.3 vs. 69.3), in comparison to OASIS without consistency regularization. CutMix-based consistency regularization only improves the mIoU (66.3 vs. 67.4),

Table B: Per-class IoU scores on ADE20k. Bold denotes the best performance.

| Classes IDs | Occupied area | mIoU | | |
|---|---|---|---|---|
| | | SPADE+ (with VGG) | OASIS without per-class balancing (without VGG) | OASIS (without VGG) |
| 0 - 29 | 86.4% | 63.7 | **69.1** | 68.8 |
| 30 - 59 | 7.2% | 47.4 | 52.4 | **56.6** |
| 60 - 89 | 3.5% | 45.3 | 47.0 | **51.5** |
| 90 - 119 | 1.8% | 29.3 | 36.2 | **41.5** |
| 120 - 149 | 1.0% | 26.2 | 31.2 | **39.7** |
| 0-149 (all classes) | 100% | 42.4 | 47.2 | **51.6** |

Table C: Per-class IoU scores on COCO-Stuff. Bold denotes the best performance.

| Classes IDs | Area | mIoU | |
|---|---|---|---|
| | | SPADE+ | OASIS |
| 0 - 35 | 69.3% | 51.1 | **59.0** |
| 36 - 69 | 15.9% | 43.9 | **50.3** |
| 70 - 103 | 8.7% | 40.5 | **40.9** |
| 104 - 137 | 4.5% | 35.9 | **36.6** |
| 138 - 171 | 1.4% | 22.1 | **40.6** |
| 0-171 (all classes) | 100% | 38.8 | **45.5** |

Table D: Per-class IoU scores on Cityscapes. Bold denotes the best performance.

| Classes IDs | Area | mIoU | |
|---|---|---|---|
| | | SPADE+ | OASIS |
| 0 - 2 | 75.6% | **91.6** | 89.6 |
| 3 - 6 | 18.3% | **75.7** | 74.9 |
| 7 - 10 | 3.9% | 60.0 | **66.9** |
| 11 - 14 | 1.4% | 60.3 | **66.0** |
| 15 - 18 | 0.6% | 38.1 | **55.1** |
| 0-18 (all classes) | 100% | 63.8 | **69.3** |

but not as much as LabelMix (69.3). We suspect that since the images are already partitioned through the label map, an additional partition through CutMix results in a dense patchwork of areas that differ by semantic class and real-fake class identity. This may introduce additional label noise during training for the discriminator. To avoid such inconsistency between semantic classes and real-fake identity, the mask of LabelMix is generated according to the label map, providing natural borders between semantic regions, so that the real and fake objects are placed side-by-side without interfering each other. Under LabelMix regularization, the generator is encouraged to respect the natural semantic class boundaries, improving pixel-level realism while also considering the class segment shapes.

## A.5 Ablation on feature matching loss

We measure the effect of the feature matching loss (FM) in the absence and presence of the perceptual loss (VGG). Table F and G present the results for OASIS on Cityscapes and SPADE+ on ADE20K. For both SPADE+ and OASIS we observe that the feature matching loss does only affect the FID notably when no perceptual loss is used.

In the case where no perceptual loss is used, we observe that the feature matching prolongs the time until SPADE+ collapses, resulting in a better FID score (49.7 vs 60.7). Consequently, the mIoU also improves. Hence, the role of the FM loss in the training of SPADE+ is to stabilize the training through additional self-supervision. This observation is in line with the general observation that SPADE and other semantic image synthesis models require the help of additional losses because the adversarial supervision through the discriminator is not strong enough. In comparison, we did not observe any training collapses in OASIS, despite not using any extra losses. For OASIS, the feature matching loss results in a worse FID (by 0.8 points) in the absence of the perceptual loss. We also observe a degradation of 1.1 mIoU points through the FM loss, in the case where the perceptual supervision is present. This indicates that the FM loss negatively affects the strong supervision from the semantic segmentation adversarial loss of OASIS.

Table E: Ablation study on the impact of LabelMix and CutMix for consistency regularization (CR) in OASIS on Cityscapes. Bold denotes the best performance.

| Transformation | FID↓ | mIoU ↑ |
|---|---|---|
| No CR | 51.5 | 66.3 |
| CutMix | 52.1 | 67.4 |
| LabelMix | **47.7** | **69.3** |

Table F: OASIS on *Cityscapes*. Bold denotes the best performance.

| VGG | FM | FID↓ | mIoU↑ |
|---|---|---|---|
| ✗ | ✗ | 47.7 | 69.3 |
| ✗ | ✓ | 48.5 | 69.1 |
| ✓ | ✗ | **46.1** | **72.0** |
| ✓ | ✓ | 46.5 | 70.9 |

Table G: SPADE+ on *ADE20K*. Bold denotes the best performance.

| VGG | FM | FID↓ | mIoU↑ |
|---|---|---|---|
| ✗ | ✗ | 60.7 | 21.0 |
| ✗ | ✓ | 49.7 | 32.5 |
| ✓ | ✗ | 32.9 | 42.5 |
| ✓ | ✓ | **32.6** | **42.9** |

## A.6 ABLATION ON USING MORE THAN ONE OASIS DISCRIMINATOR

A major difference between SPADE and OASIS is that OASIS employs only one discriminator, while SPADE uses two PatchGAN discriminators at different scales. Naturally, the question arises how OASIS performs with two discriminators at different scales, as in SPADE. For this, Table H presents the FID and mIoU performance of OASIS with two discriminators operating at scales 1 and 0.5 on Cityscapes. One can see that an additional discriminator at scale 0.5 does not improve performance, but slightly worsens it. The reason that no performance gain is visible is that the OASIS discriminator already encodes multi-scale information through its U-Net structure: skip connections between encoder, decoder and individual blocks integrate information at all scales. In contrast, SPADE requires two discriminators to capture information at different scales.

Table H: Comparison of using 1 and 2 discriminators at different scales for OASIS on Cityscapes. Bold denotes the best performance.

| # of OASIS D | FID↓ | mIoU↑ |
|---|---|---|
| 1 discriminator | **47.7** | **69.3** |
| 2 discriminators at different scales (1 and 0.5) | 48.7 | 68.8 |

## A.7 ABLATION ON NOISE SAMPLING STRATEGIES DURING TRAINING

Our 3D noise can contain the same sampled vector for each pixel, or different vectors for different regions. This allows for different noise sampling schemes during training. Table I shows the effect of using different methods of sampling 3D noise for different locations during training: **Image-level** sampling creates one global 1D noise vector and replicates it along the height and width of the label map to create a 3D noise tensor. **Region-level** sampling relies on generating one 1D noise vector per label, and stacking them in 3D to match the height and width of the label map. **Pixel-level** sampling creates different noise for every spatial position, with no replication taking place. **Mix** switches between image-level and region-level sampling via a coin flip decision at every training step. With no obvious winner in performance, we choose the simplest scheme (image-level) for our experiments.

By choosing image-level sampling for training, we thus generate a single 1D latent noise vector of size 64, broadcast it to 64xHxW and concatenate with the label map (NxHxW). This new composite tensor is used as input to the 1st generator layer and at all SPADE-norm layers. The noise is not ignored for the following reasons:

**(1)** The noise modulates the activations directly at *every* layer, so it is very hard to ignore. Here, it is important to emphasize how the noise is used: For SPADE it was observed that label maps are paid more attention to if they are used for location-sensitive conditional batch normalization (CBN). Analogously, we observe that the noise is also paid more attention to when it is injected via CBN. Like label maps, which are 3D tensors of stacked one-hot vectors, we stack noise vectors into a 3D

tensor of the same dimensions. Thus, in the same way that SPADE is spatially sensitive to labels, OASIS is spatially sensitive to both labels and noise.

**(2)** The 3D broadcasting strategy provides a spatially uniform signal making it easy to embed semantic meaning into the latent code (see interpolations, Fig. I , J). As noise modulates features at different scales in the generator, matching their resolution, it affects both global and local characteristics. This is why a generator trained with image-level noise can perform region-level manipulation at inference (Fig. F, H). However, more evolved spatial noise sampling schemes can be explored in the future.

Table I: Different noise sampling strategies during training. Bold denotes the best performance.

| Sampling | Cityscapes | | | ADE20K | | |
|---|---|---|---|---|---|---|
| | FID↓ | mIoU↑ | MS-SSIM↓ | FID↓ | mIoU↑ | MS-SSIM↓ |
| image-level | 47.7 | 69.3 | 0.64 | **28.3** | **48.8** | 0.65 |
| region-level | 48.1 | 69.7 | **0.62** | 28.8 | 48.1 | **0.58** |
| pixel-level | 50.9 | 65.5 | 0.84 | 28.6 | 34.0 | 0.68 |
| mix | **46.4** | **70.9** | 0.68 | 28.5 | 47.6 | 0.66 |

## A.8 ADDITIONAL EXPERIMENTS ON COCO-STUFF

We performed all our extensive ablations on ADE20K and Cityscapes, due to their shorter training time. Training on ADE20K and Cityscapes takes circa 10 days on 4 Tesla V100 GPUs while training on COCO-stuff can stretch to 4 weeks. Therefore, we only executed essential experiments on COCO-stuff. We compare the results of these experiments in Table J. For SPADE+, it can be seen that without the external perceptual supervision of VGG, training collapses (with FID 99.1 at the best checkpoint before collapse). In contrast, for OASIS image quality is better without VGG (16.7 vs 18.0 FID).

Table J: Performance on COCO-stuff. Bold denotes the best perfromance.

| Model | VGG | 3D noise | FID↓ | mIoU↑ | MS-SSIM↓ |
|---|---|---|---|---|---|
| SPADE | ✓ | ✗ | 22.6 | 37.4 | 1.0 |
| SPADE+ | ✗ | ✗ | 99.1 | 16.1 | 1.0 |
| SPADE+ | ✓ | ✗ | 21.7 | 38.8 | 1.0 |
| OASIS | ✗ | ✗ | **16.7** | **45.5** | 1.0 |
| OASIS | ✓ | ✗ | 18.0 | 44.2 | 1.0 |
| OASIS | ✗ | ✓ | 17.0 | 44.1 | **0.61** |

When 3D noise is added to OASIS, sampling of multi-modal images is enabled (0.61 vs 1.0 MS-SSIM), with very similar performance in synthesis quality (17.0 vs 16.7 FID) and slightly worse mIoU (44.1 vs 45.5 mIoU) due to the increased variation of generated samples, as the semantic segmentation task of the pre-trained segmentation network becomes harder.

## A.9 ADDITIONAL EVALUATION METRICS

Currently, the FID score is the most widely adopted metric for quantifying image quality of GAN models. However, it is often argued that the FID score does not adequately disentangle synthesis quality and diversity (Kynkäänniemi et al., 2019). Recently, a series of metrics have been proposed to address this issue by measuring scores related to the concepts of precision and recall (Ravuri & Vinyals, 2019; Shmelkov et al., 2018; Sajjadi et al., 2018; Kynkäänniemi et al., 2019). Here, we have a closer look at the "improved precision and recall" score proposed by (Kynkäänniemi et al., 2019), where precision is the probability that a generated image falls into the estimated support of the real image distribution, and recall is the probability that a real image falls into the estimated support of the generator distribution. Precision and recall can be interpreted as sample quality and diversity. Table K presents a comparison of precision (P) and recall R) between SPADE+ and OASIS. It can be seen that OASIS outperforms SPADE+ both in terms of image quality and variety.

Table K: Comparison of the precision and recall metric between SPADE+ and OASIS. Bold denotes the best performance.

| Model | ADE20K | | ADE-outd. | | Cityscapes | | COCO-Stuff | |
|---|---|---|---|---|---|---|---|---|
| | P↑ | R↑ | P↑ | R↑ | P↑ | R↑ | P↑ | R↑ |
| SPADE+ | 0.71 | 0.52 | 0.62 | 0.51 | 0.54 | 0.34 | 0.63 | 0.56 |
| OASIS | **0.77** | **0.57** | **0.77** | **0.56** | **0.58** | **0.55** | **0.67** | **0.59** |

# B  QUALITATIVE RESULTS

## B.1  COMPARISON TO OTHER METHODS

In this section we present a visual comparison between OASIS and other semantic image synthesis methods. Firstly, we show images generated by SPADE (Park et al., 2019), CC-FPSE (Liu et al., 2019) and OASIS on ADE20k, COCO-Stuff, and Cityscapes (in Figures B, C, and D, respectively). A further comparison for SPADE, SPADE+ and OASIS is presented in Figure E. We observed that OASIS often produces more visually plausible images than the previous methods. Our method commonly produces finer textures, especially for complex and large semantic objects, e.g building facades, mountains, water.

We also note that OASIS usually generates brighter and more diverse colors, compared to other methods. As we showed in Section 4 in the main paper, the diversity in colors partially comes from the fact that the feature space of the OASIS generator is not constrained by the VGG loss. We observed that images, generated by SPADE and CC-FPSE, typically have closer colors, while OASIS frequently generates objects with completely different color tones. This also forms one of the failure modes of our approach, when the colors of objects fall out of distribution and seem unnatural (see Figure G).

## B.2  MULTI-MODAL IMAGE SYNTHESIS

Multi-modal image synthesis for a given label map is easy for OASIS: we simply re-sample noise like in a conventional unconditional GAN model. Since OASIS employs a 3D noise tensor (64-channels×height×width), the noise can be re-sampled entirely ("globally") or only for specific regions in the 2D image plane ("locally"). For our visualizations, we replicate a single 64-dimensional noise vector along the spatial dimensions for global sampling. For local sampling, we re-sample a new noise vector and use it to replace the global noise vector at every spatial position within a restricted area of interest. The results are shown in Figure F. The generated images are diverse and of high quality. We observe different degrees of variety for different object classes. For example, buildings change drastically in appearance and often change their spatial orientation with respect to the road. On the other side, many common objects (like tables) vary in color, texture, and illumination, but do not change shapes as they are restricted by the fine details of the region that is outlined for them in the label map.

Local noise re-sampling does not have to be restricted to only semantic class areas: in Figure H we sample a different noise vector for the left and right half of the image, as well as for arbitrarily shaped regions. In effect, the two areas can differ substantially. However, often a bridging element is found between two areas, such as clouds extending partly from one region to the other region of the image.

## B.3  LATENT SPACE INTERPOLATIONS

In Figure I we present images that are the results of linear interpolations in the latent space (see Fig. I), using an OASIS model trained on the ADE20K dataset. To generate the images, we sample two noise vectors $z \in \mathbb{R}^{64}$ and interpolate them with three intermediate points. The images are synthesized for these five different noise inputs while the label map is held fixed. Note that in Figure I we only vary the noise *globally*, not locally (See Section 3.3 in the main paper). In contrast, Figure J shows *local* interpolations. For this, we only re-sample the 3D noise in the area corresponding to a single semantic class. The effect is that only the appearance of the selected semantic class varies while the rest of the image remains fixed. It can be observed that strong changes in a local area can slightly affect the surroundings if the local area is also very big. As such, the clouds are slightly different in the first and last panel of the mountain row and tree row in Figure J.

We see from Figure I and J that the trajectories in the latent space are smooth and semantically meaningful. For example, we observe transitions from winter to summer, day to night, green trees to leafless trees, shiny parquet to matt carpet, as well as smooth transitions between buildings with different architectural styles.

### B.4 APPLICATION TO UNLABELLED DATA

OASIS has a unique property that its discriminator is trained to be an image segmenter. We observed that it shows good performance on this task, reaching the mIoU of 40.0 on ADE20K validation set. For comparison, current state of the art on ADE20K is a mIoU of 46.91, achieved by ResNeST (Zhang et al., 2020). Such a good segmentation performance allows OASIS to be applied to unlabelled images: given an unseen image without a ground truth annotation, OASIS can predict a label map via the discriminator. Subsequently feeding this prediction to the generator allows to synthesize a scene with the same layout but different style. This property is shown in Fig. K. Due to the good segmentation performance, the recreated scenes closely follow the ground truth label map of the original image. The high sensitivity of OASIS to the 3D noise enforces good variability, so the recreations are different from each other. We believe that creating multiple versions of one image while retaining the layout can be useful for data-augmentation.

### B.5 LABELMIX

Figure L shows additional visual examples of LabelMix regularization, as described in Section 3.2 in the main paper. It can be seen that the discriminator prediction on the mixed images often differs from the mix of individual predictions on real and fake images. In particular, regions that are classified as real in the latter are classified as fake when the images are mixed. This means that the discriminator takes the global context into account for local predictions and thereby often bases the prediction on arbitrary details that should not affect the real-fake class identity. In return, the consistency regularization helps to minimize the difference between these two predictions.

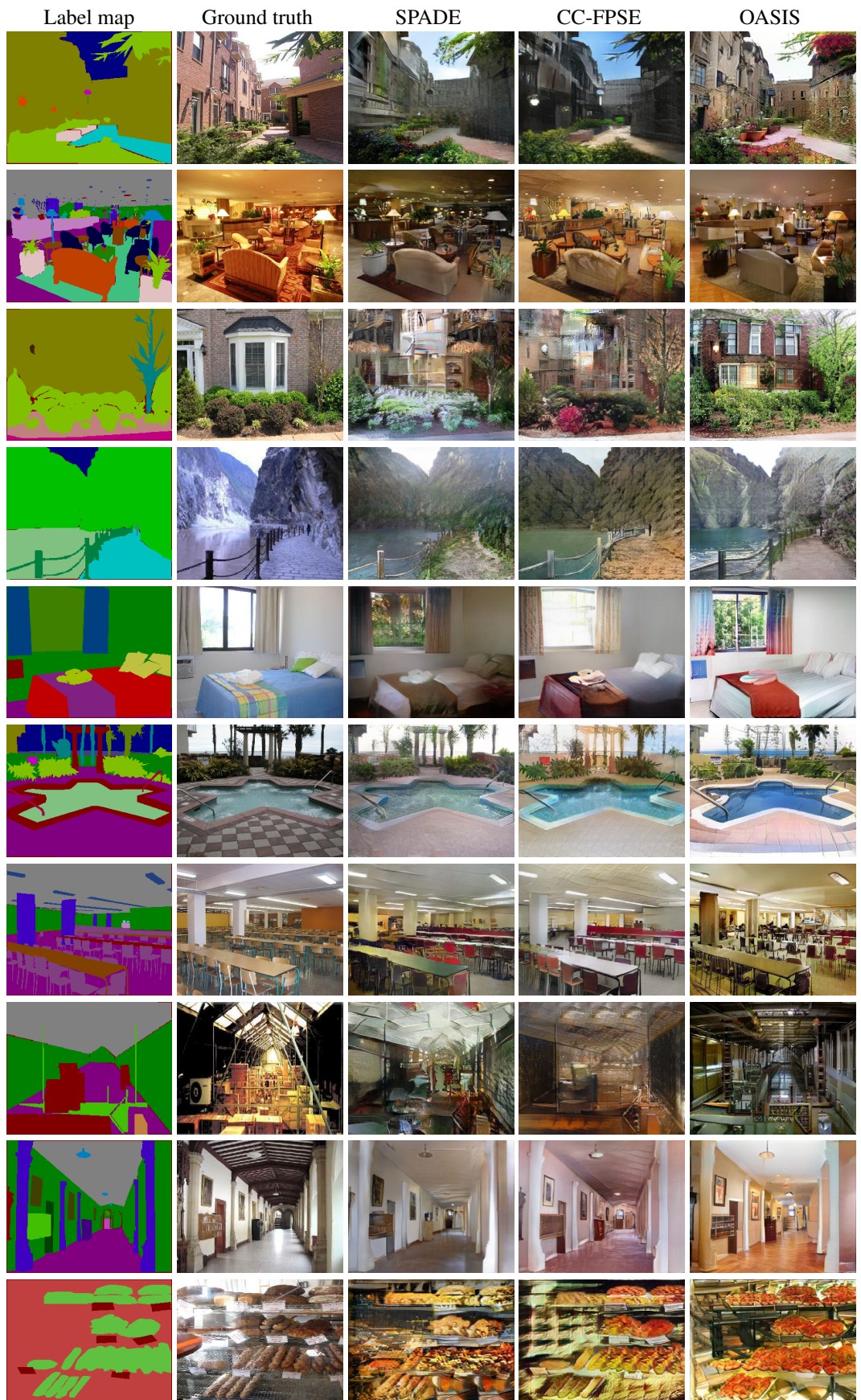

Label map    Ground truth    SPADE    CC-FPSE    OASIS

Figure B: Qualitative comparison of OASIS with other methods on ADE20K.

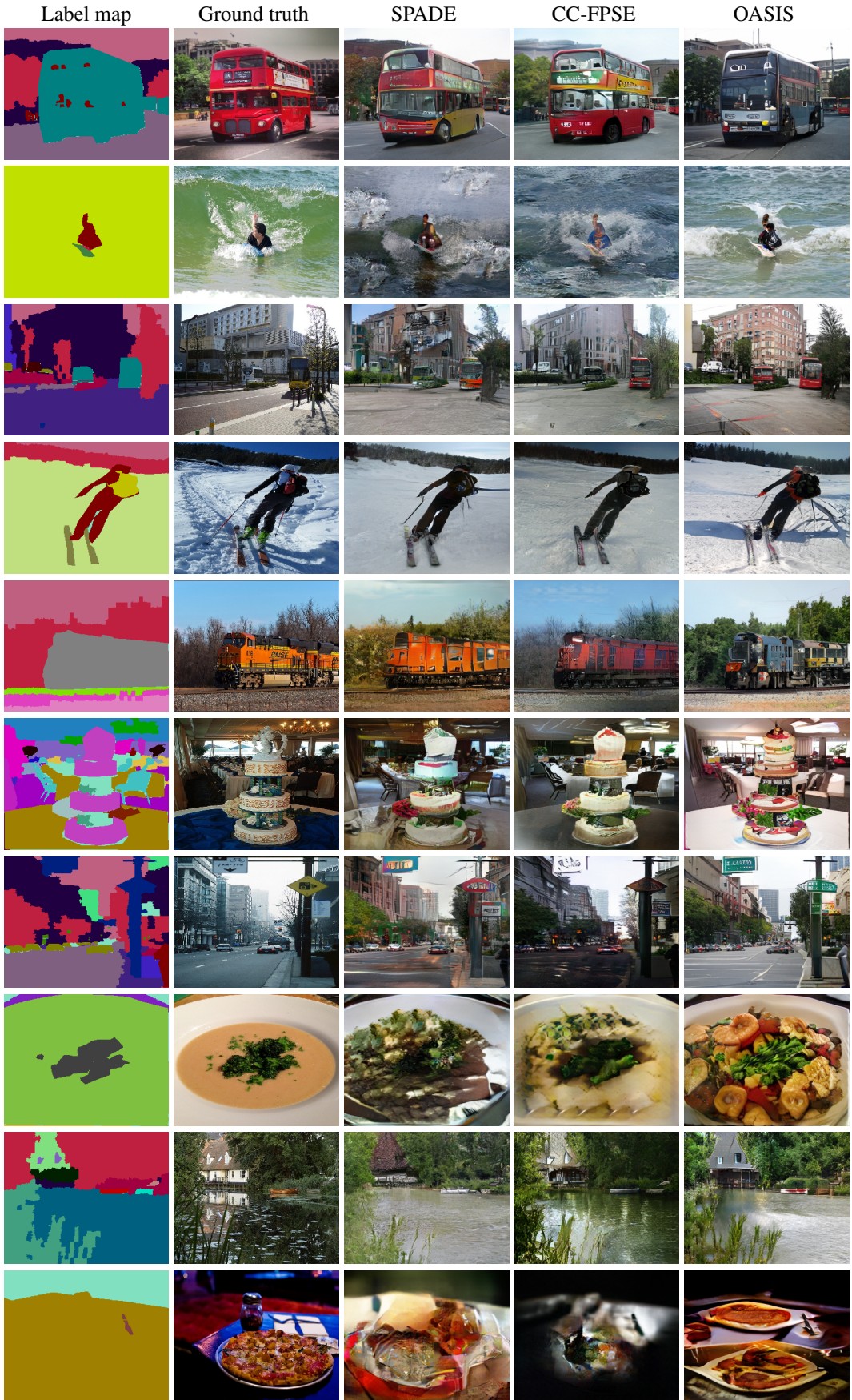

Figure C: Qualitative comparison of OASIS with other methods on COCO-Stuff.

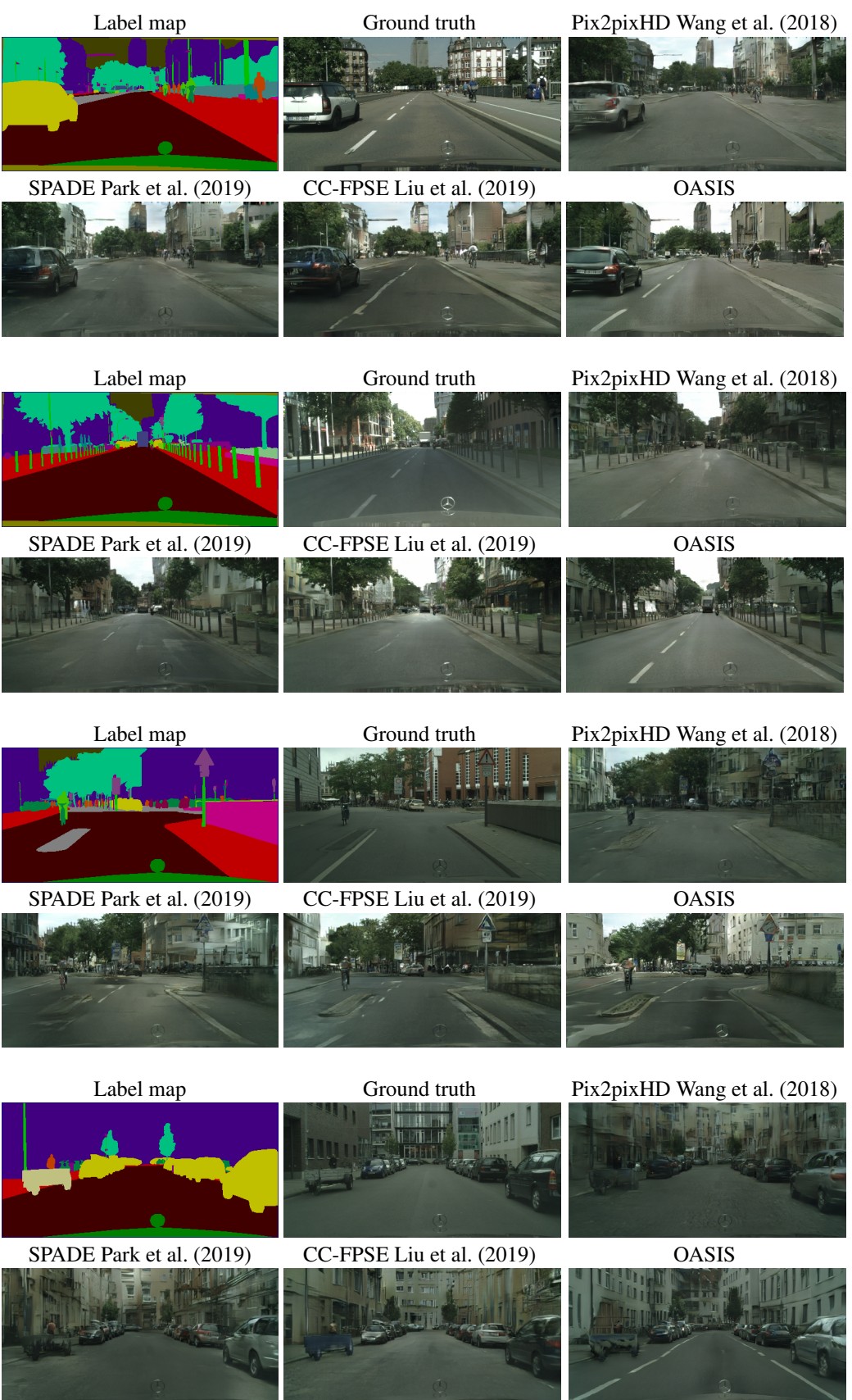

Figure D: Qualitative comparison of OASIS with other methods on Cityscapes.

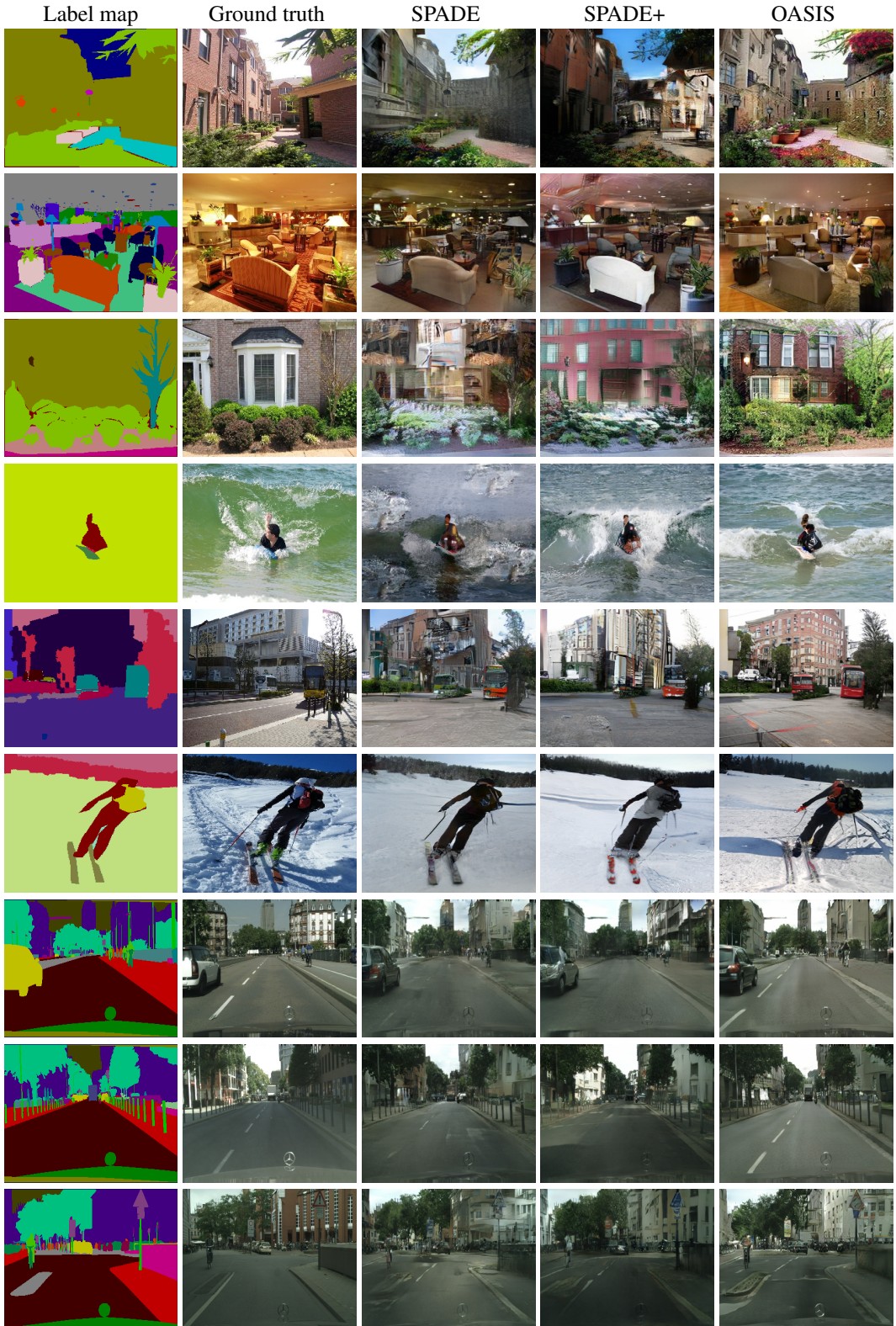

Figure E: Qualitative comparison of OASIS with SPADE and SPADE+ using ADE20K (row 1-3), COCO-stuff (row 4-6) and Cityscapes (row 7-9).

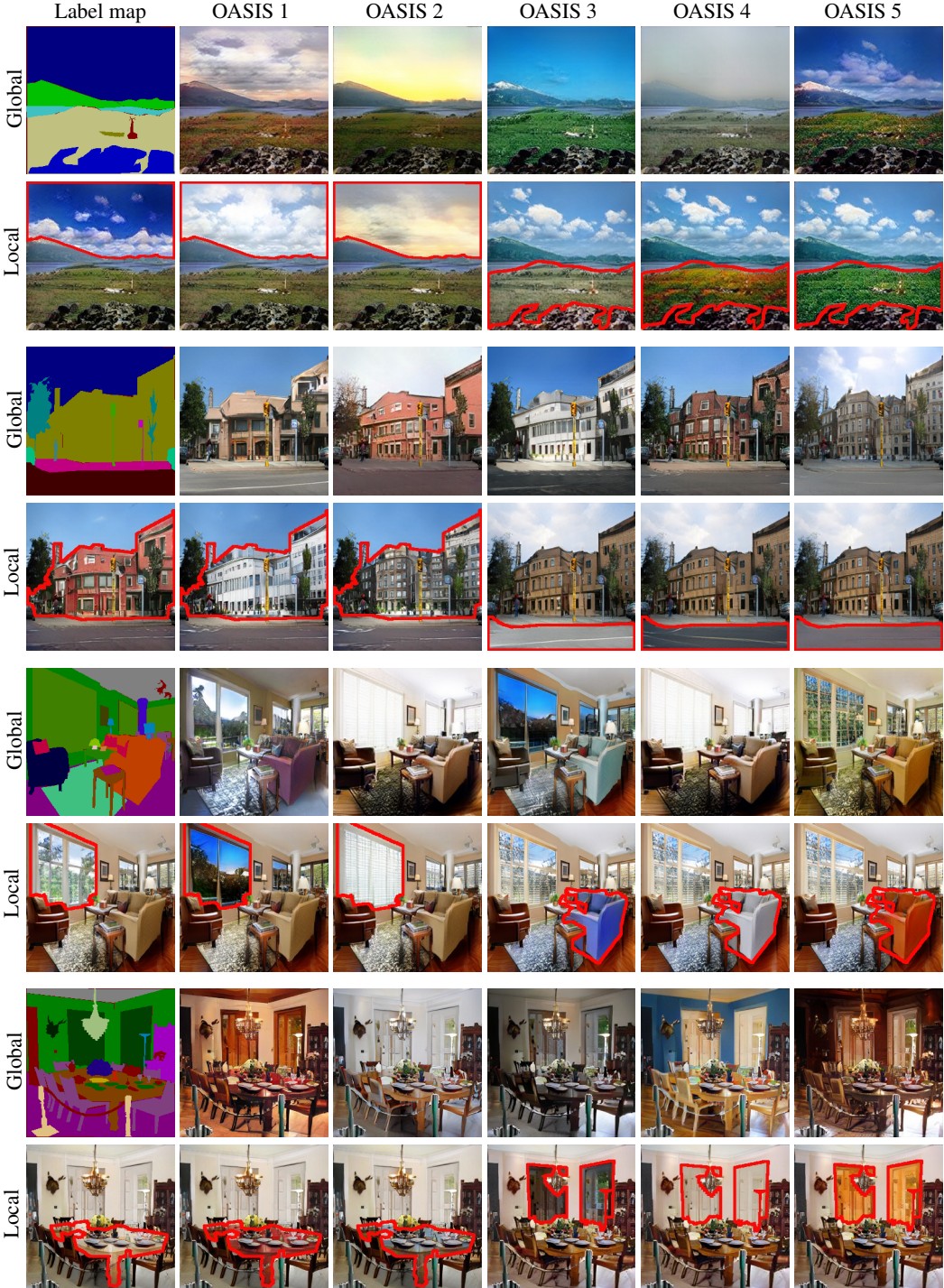

Figure F: Images generated by OASIS on ADE20K with $256 \times 256$ resolution using different 3D noise inputs. For each label map the noise is re-sampled globally (first row) or locally in the areas marked in red (second row). Note that the images are not stitched together but generated in single forward passes.

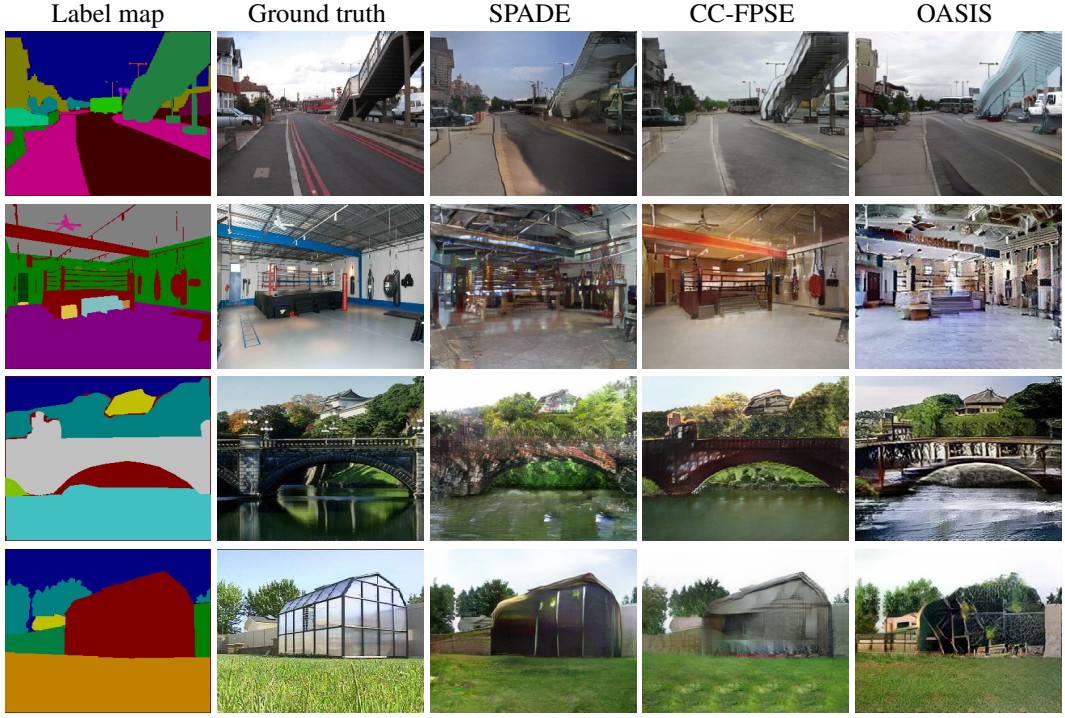

Figure G: Failure mode of OASIS. Our model generates diverse images, sometimes producing object with outlier colors and textures. We compare to Park et al. (2019) and Liu et al. (2019).

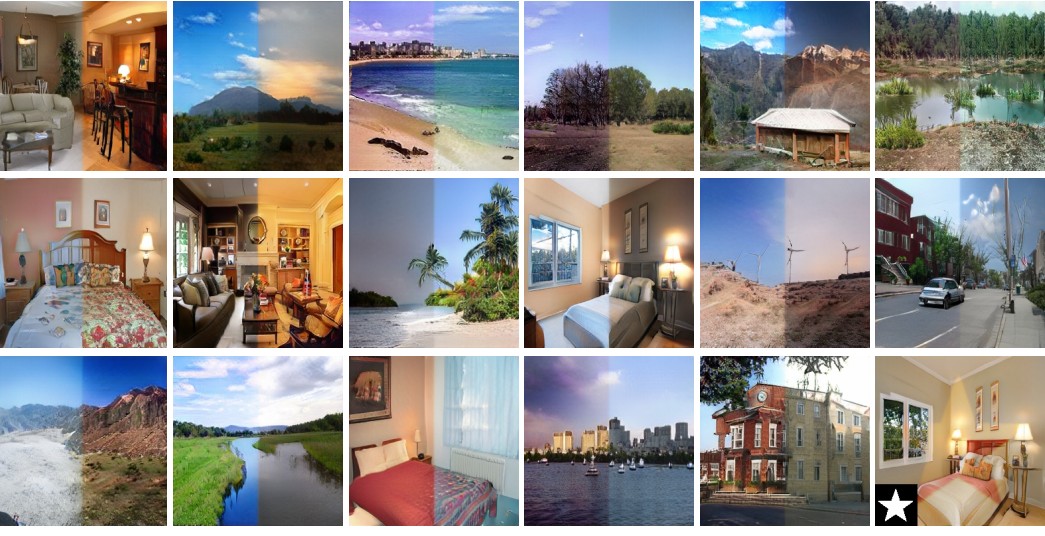

Figure H: Images generated by OASIS *in one forward pass* (no collage), with different noise vectors for different image regions.

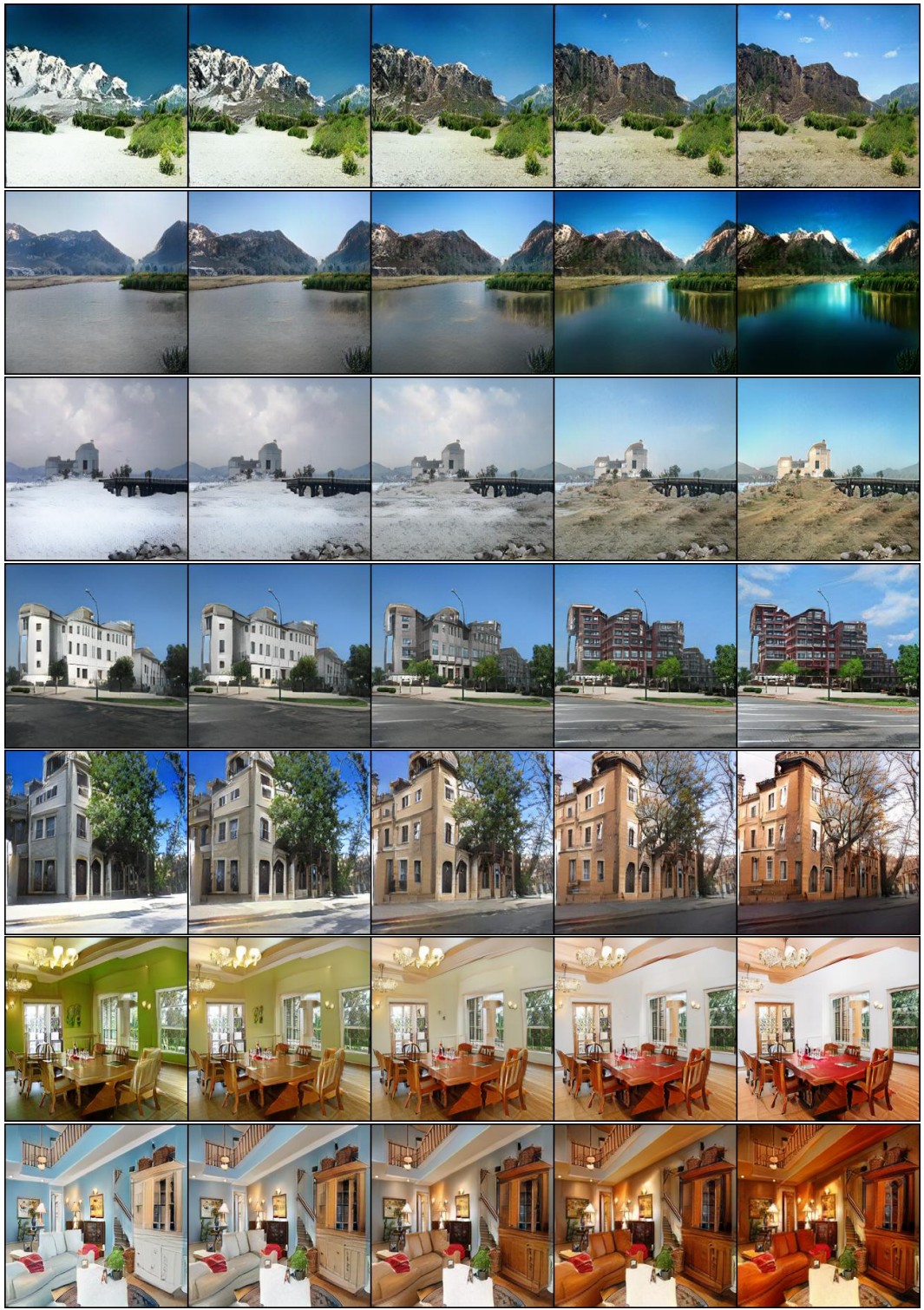

Figure I: *Global* latent space interpolations between images generated by OASIS for various outdoor and indoor scenes in the ADE20K dataset at resolution $256 \times 256$.

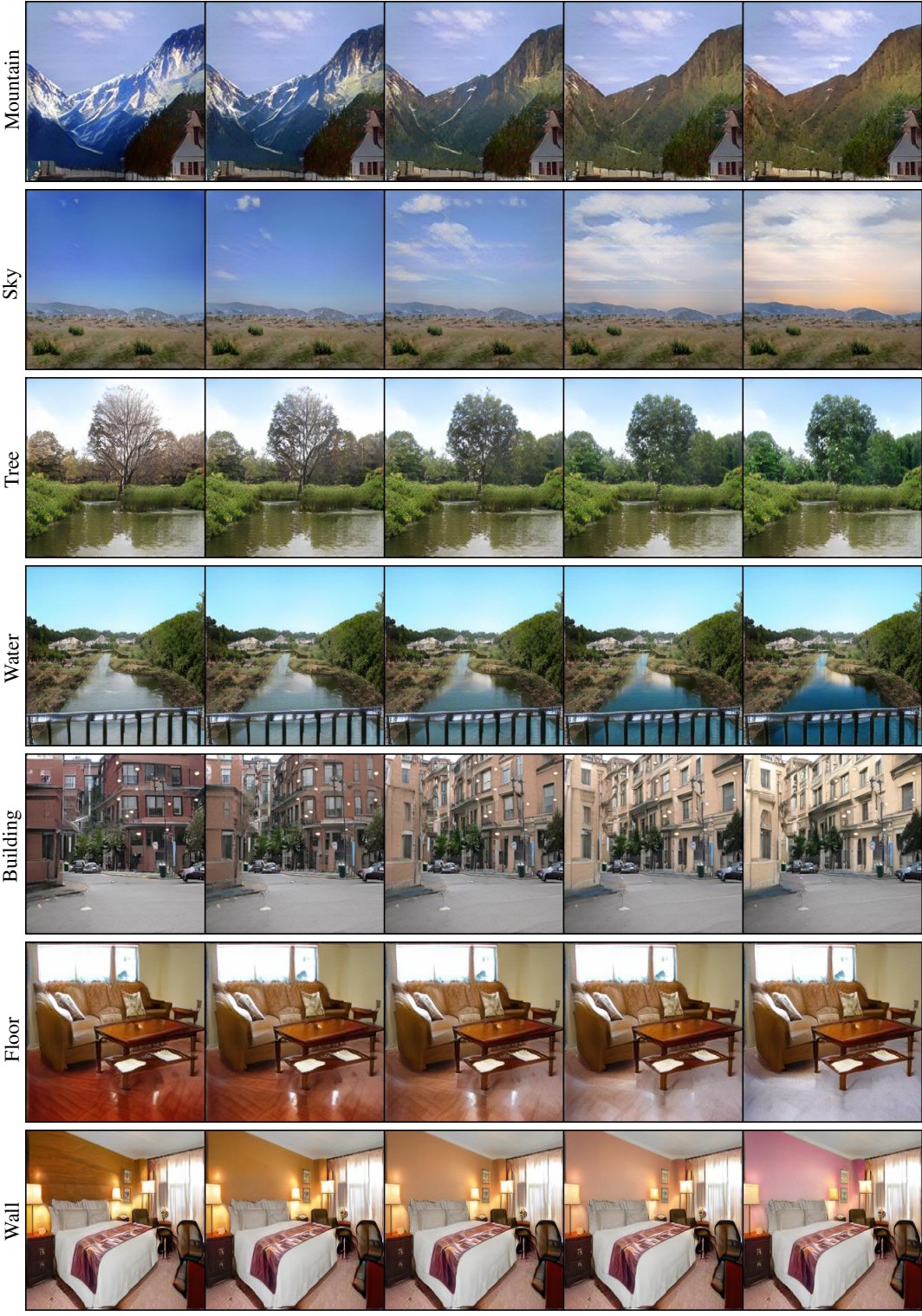

Figure J: Latent space interpolations in *local* regions of the 3D noise, corresponding to a single semantic class. The noise is only changed within the restricted area. Trained on the ADE20K dataset at resolution $256 \times 256$.

GT label map    Input image    Segmentation    Recreation 1    Recreation 2    Recreation 3

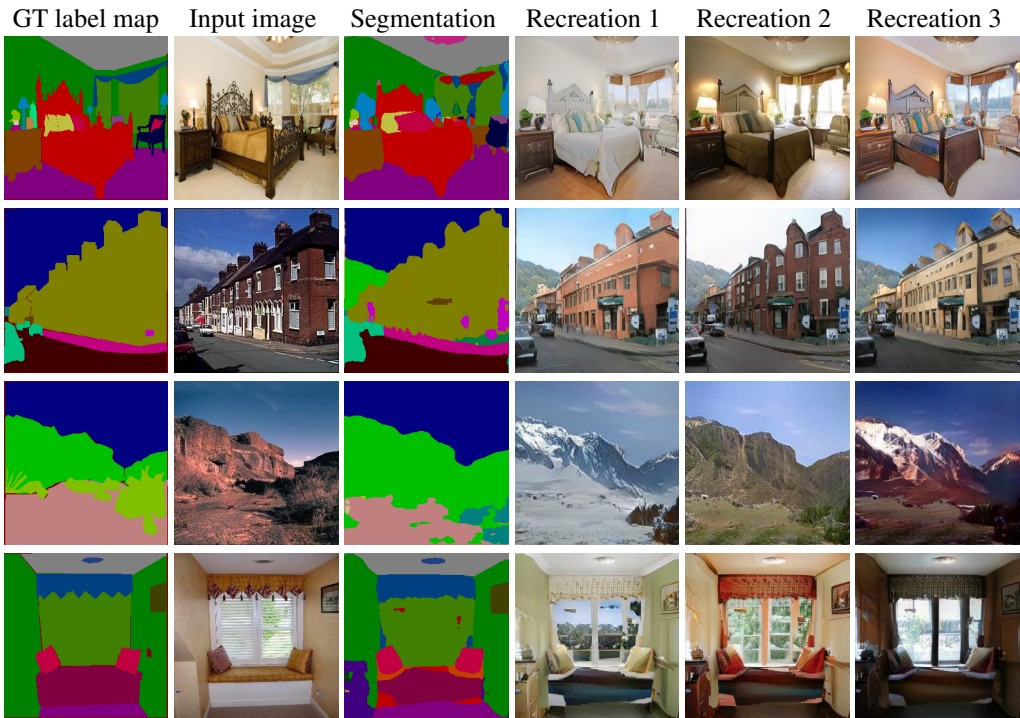

Figure K: After training, the OASIS discriminator can be used to segment images. Columns 1-3 show the ground truth label map, real image, and segmentation of the discriminator. Using the predicted label map the generator can produce multiple versions of the original image by resampling noise (Recreations 1-3). Note that this alleviates the need of ground truth maps during inference.

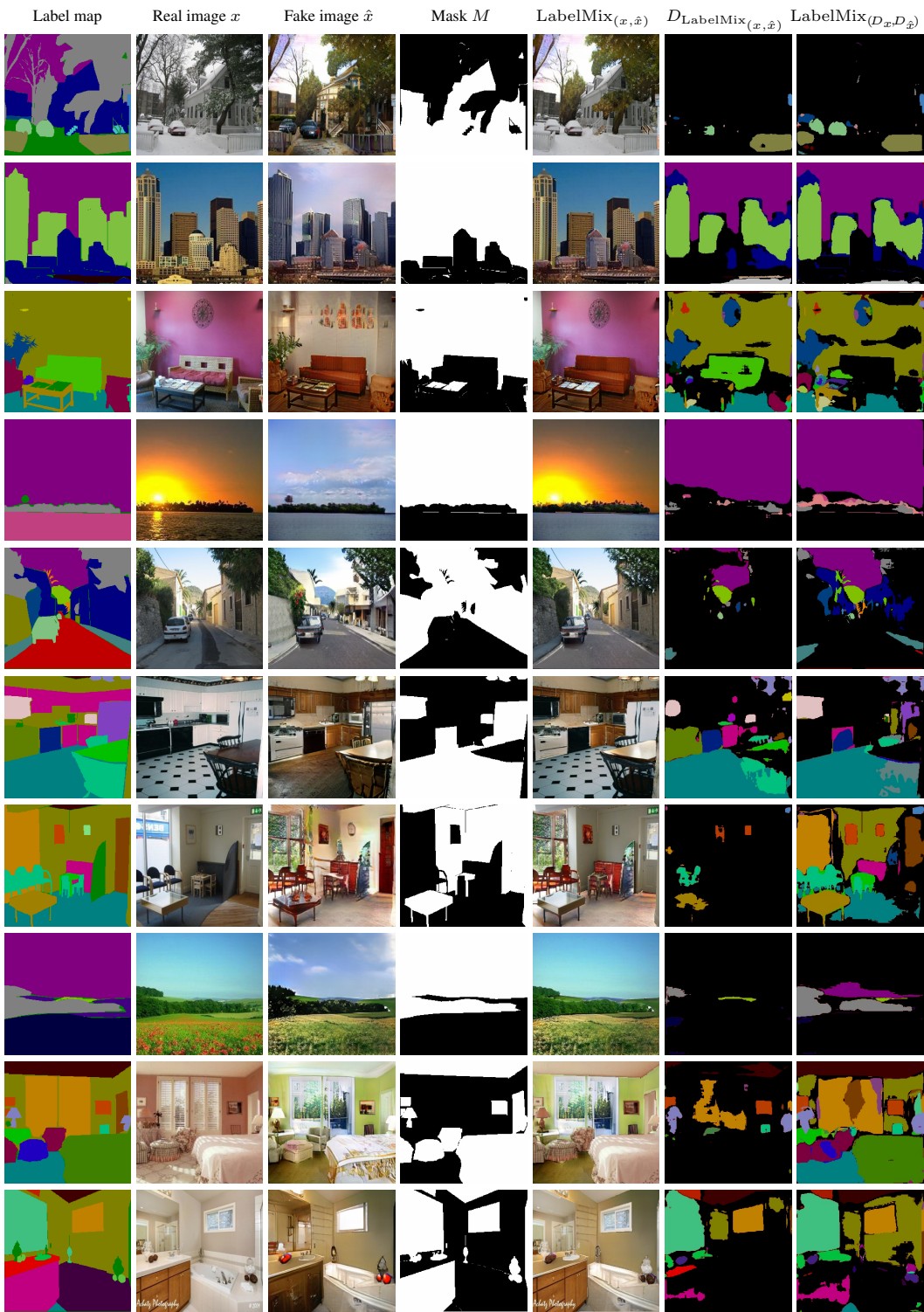

Figure L: Visual examples of LabelMix regularization. Real $x$ and fake $\hat{x}$ images are mixed using a binary mask $M$, sampled based on the label map, resulting in $\text{LabelMix}_{(x,\hat{x})}$. The consistency regularization then minimizes the distance between the logits of $D_{\text{LabelMix}_{(x,\hat{x})}}$ and $\text{LabelMix}_{(D_x,D_{\hat{x}})}$. In this visualization, **black** corresponds to the fake class in the $N+1$ segmentation output.

## C  ARCHITECTURAL AND TRAINING DETAILS

The architecture of OASIS builds upon SPADE Park et al. (2019). In the following, we describe in detail our proposed changes to the discriminator and the generator.

### C.1  DISCRIMINATOR ARCHITECTURE

This OASIS discriminator follows a U-Net architecture and is built from ResNet blocks, inspired in their design by Brock et al. (2019). The architecture of the OASIS discriminator is outlined in Table L. It has in total 22M learnable parameters and is bigger than the multi-scale PatchGAN discriminator (5.5M) used by SPADE Park et al. (2019). The increased capacity of the OASIS discriminator allows it to learn a more powerful representation and provide more informative feedback to the generator.

Table L: The OASIS discriminator. N refers to the number of semantic classes.

| Operation | Input | Size | Output | Size |
|---|---|---|---|---|
| ResBlock-Down | `image` | (3,256,256) | `down_1` | (128,128,128) |
| ResBlock-Down | `down_1` | (128,128,128) | `down_2` | (128,64,64) |
| ResBlock-Down | `down_2` | (128,64,64) | `down_3` | (256,32,32) |
| ResBlock-Down | `down_3` | (256,32,32) | `down_4` | (256,16,16) |
| ResBlock-Down | `down_4` | (256,16,16) | `down_5` | (512,8,8) |
| ResBlock-Down | `down_5` | (512,8,8) | `down_6` | (512,4,4) |
| ResBlock-Up | `down_6` | (512,4,4) | `up_1` | (512,8,8) |
| ResBlock-Up | `cat(up_1, down_5)` | (1024,8,8) | `up_2` | (256,16,16) |
| ResBlock-Up | `cat(up_2, down_4)` | (512,16,16) | `up_3` | (256,32,32) |
| ResBlock-Up | `cat(up_3, down_3)` | (512,32,32) | `up_4` | (128,64,64) |
| ResBlock-Up | `cat(up_4, down_2)` | (256,64,64) | `up_5` | (128,128,128) |
| ResBlock-Up | `cat(up_5, down_1)` | (256,128,128) | `up_6` | (64,256,256) |
| Conv2D | `up_6` | (64,256,256) | `out` | (N+1,256,256) |

### C.2  GENERATOR ARCHITECTURE

The generator architecture is built from SPADE ResNet blocks and includes a concatenation of 3D noise with the label map along the channel dimension as an option. The generator can be either trained directly on the label maps or with 3D noise concatenated to the label maps. The latter option is shown in Table M.

OASIS generator drops the first residual block used in Park et al. (2019), which decreases the number of learnable parameters from 96M to 72M. The optional 3D noise injection brings additionally 2M parameters. This sampling scheme is five times lighter than the image encoder used by SPADE (10M).

### C.3  LEARNING OBJECTIVE AND TRAINING DETAILS

**Learning objective.** We train our model with $(N+1)$-class cross entropy as an adversarial loss. Additionally, the discriminator is regularized with the proposed LabelMix consistency regularization. The full OASIS learning objective thus takes the following form:

$$\mathcal{L}_G^{\text{OASIS}} = -\mathbb{E}_{(z,t)}\left[\sum_{c=1}^{N}\alpha_c\sum_{i,j}^{H\times W}t_{i,j,c}\log D(G(z,t))_{i,j,c}\right],$$

$$\mathcal{L}_D^{\text{OASIS}} = -\mathbb{E}_{(x,t)}\left[\sum_{c=1}^{N}\alpha_c\sum_{i,j}^{H\times W}t_{i,j,c}\log D(x)_{i,j,c}\right] - \mathbb{E}_{(z,t)}\left[\sum_{i,j}^{H\times W}\log D(G(z,t))_{i,j,c=N+1}\right] +$$

$$+ \lambda_{\text{LM}}\left\|D_{\text{logits}}\Big(\text{LabelMix}(x,\hat{x},M)\Big) - \text{LabelMix}\Big(D_{\text{logits}}(x), D_{\text{logits}}(\hat{x}), M\Big)\right\|_2^2,$$

where $x$ denotes the real image and $(z,t)$ is the noise-label map.

Table M: The OASIS generator. N refers to the number of semantic classes, z is noise sampled from a unit Gaussian, y is the label map, interp interpolates a given input to the appropriate spatial dimensions of the current layer.

| Operation | Input | Size | Output | Size |
|---|---|---|---|---|
| Concatenate | z_3D
y | (64,256,256)
(N,256,256) | z_y | (64+N,256,256) |
| Conv2D | interp(z_y) | (64+N,8,8) | x | (1024,8,8) |
| SPADE-ResBlock | x
interp(z_y) | (1024,8,8)
(64+N,8,8) | up_1 | (1024,16,16) |
| SPADE-ResBlock | up_1
interp(z_y) | (1024,16,16)
(64+N,16,16) | up_2 | (512,32,32) |
| SPADE-ResBlock | up_2
interp(z_y) | (512,32,32)
(64+N,32,32) | up_3 | (256,64,64) |
| SPADE-ResBlock | up_3
interp(z_y) | (256,64,64)
(64+N,64,64) | up_4 | (128,128,128) |
| SPADE-ResBlock | up_4
interp(z_y) | (128,128,128)
(64+N,128,128) | up_5 | (64,256,256) |
| Conv2D, LeakyRelu, TanH | up_5 | (64,256,256) | x | (3,256,256) |

Our objective function is different from SPADE. Their model uses hinge adversarial loss and adds the VGG perceptual loss and a feature matching loss to train the generator. For an easier comparison, we provide the objective function of SPADE:

$$\mathcal{L}_G^{\text{SPADE}} = -\mathbb{E}_{(z,t)} \left[ D(t, G(z,t)) \right] + \lambda_{\text{FM}} \, \mathbb{E}_{(z,t,x)} \sum_{i=1}^{T} \| D_k^{(i)}(t,x) - D_k^{(i)}(t, G(z,t)) \|_1 +$$

$$+ \lambda_{\text{VGG}} \mathbb{E}_{(z,t,x)} \sum_{i=1}^{N} \| F^{(i)}(x) - F^{(i)}(G(z,t)) \|_1,$$

$$\mathcal{L}_D^{\text{SPADE}} = -\mathbb{E}_{(t,x)} \left[ \min(0, -1 + D(t,x)) \right] - \mathbb{E}_{(z,t)} \left[ \min(0, -1 - \log D(t, G(z,t))) \right],$$

where $F$ is the pre-trained VGG network.

**Training details.** We follow the experimental setting of (Park et al., 2019). The image resolution is set to 256x256 for ADE20K and COCO-Stuff and 256x512 for Cityscapes. The Adam (Kingma & Ba, 2015) optimizer was used with momentums $\beta = (0, 0.999)$ and constant learning rates $(0.0001, 0.0004)$ for $G$ and $D$. We did not apply the GAN feature matching loss, and used the VGG perceptual loss only for ablations with $\lambda_{\text{VGG}} = 10$. The coefficient for LabelMix $\lambda_{\text{LM}}$ was set to 5 for ADE20k and Cityscapes, and to 10 for COCO-Stuff. All our models use an exponential moving average (EMA) of the generator weights with 0.9999 decay (Brock et al., 2019). All the experiments were run on 4 Tesla V100 GPUs, with a batch size of 20 for Cityscapes, and 32 for ADE20k and COCO-Stuff. The training epochs are 200 on ADE20K and Cityscapes, and 100 for the larger COCO-Stuff dataset. On average, a complete forward-backward pass with batch size 32 on Ade20k takes around 0.95ms per training image.

