# OpenReview forum: "You Only Need Adversarial Supervision for Semantic Image Synthesis"
_ICLR.cc/2021/Conference — ICLR 2021 Poster_

### Official Review · AnonReviewer4 · 2020-10-28
**Excellent improvement of SPADE+. Would like to see ablations on COCO and potentially results on Open Images.**

**Rating:** 7
**Confidence:** 5

**Review:**

# Post Rebuttal:

Thank you for the detailed rebuttal. The comment made about being able to use this in a semi-supervised setting is an exciting direction and I encourage the authors to pursue it on larger less-labeled datasets mentioned in the review in a future work/final submission. I am glad that removing VGG improved the results on COCO.

Ultimately, I am keeping my score at 7, accept.

# Pre-Rebuttal:

Summary
=============================================================================================================

This paper provides an improvement to the SPADE+ architecture that removes the need for a perceptual loss mechanism while also quantitatively improving the outputs with respect to FID/mmIOU. The paper achieves this while reducing the number of params and removing the ImageNet bias of the original perceptual VGG architecture. The authors evaluate their architecture on 3 datasets: ADE20K, Cityscapes and COCO-stuff. They do while using only adversarial supervision. They achieve this by proposing label mix to train the network by having the discriminator output class labels including an additional "fake" label in addition to a variety of other optimizations.

Overall, I vote for accepting this paper. It significantly improves on the SPADE+ architecture by removing a pre-trained component and reducing the number of params. The technique seems to improve upon prior work qualitatively, quantitatively, and performance-wise.

Pros
==============================================================================================================
1. The paper is very well written.
1. The paper does provide a very thorough comparison with prior works.
1. This paper improves on the results of prior methods while reducing the # parameters and the need for imagenet pretraining.

Cons:
==============================================================================================================
1. Pixel level semantic segmentations are difficult and expensive to generate. This limits the number of datasets that can be used for these work. In this sense, the data is highly supervised.
1. That being said, I would love to the results on a more challenging dataset like LVIS and/or Open Images. For Open Images, it would be interesting to see if the performance scales with 350+ classes. For the former, it would be interesting to see how well it does on objects that appear very rarely. (I don't expect the results on LVIS to be good, but I would expect them to be better than SPADE+).
1. Architectural ablations only cover the first two datasets, any reasons why there are none for COCO-stuff?  One of the main claims of the paper, (that this technique does better without VGG), is not evaluated on COCO-stuff.

---

> ### Author Response · Authors · 2020-11-18
> **Response to Reviewer 4**
>
> **R4-3:** *“Architectural ablations only cover the first two datasets, any reasons why there are none for COCO-stuff? One of the main claims of the paper, (that this technique does better without VGG), is not evaluated on COCO-stuff.”*
>
> We conducted our ablations on ADE20k and Cityscapes due to their shorter training time. Training models on COCO-stuff can take up to 4 weeks, while on ADE20K and Cityscapes it takes circa 10 days. Therefore, we only executed for a few ablations on COCO-stuff. In particular, we conducted experiments where OASIS is trained in the absence of 3D noise, with and without VGG. The results are as follows: With VGG, FID and mIoU are 18.0 and 44.2. Without VGG, FID and mIoU are 16.7 and 45.5. Thus, OASIS without VGG performs better. For comparison, our final OASIS model (trained without VGG but with noise) achieves FID of 17.0 and mIoU of 44.1, with the added benefit of generating multi-modal images via the 3D noise sampling.
>
> Lastly, training SPADE+ without VGG leads to collapse, which once more validated the dependence of SPADE+ on VGG.  The above results are aligned with our findings on ADE20K and Cityscapes. We present these additional results on COCO-stuff in Appendix A8 and Table J.

---

> ### Author Response · Authors · 2020-11-18
> **Response to Reviewer 4**
>
> Thank you for the feedback and the valuable suggestions concerning the additional big datasets. Let us next address the points you raised:
>
> $ $
>
> **R4-1:** *“Pixel level semantic segmentations are difficult and expensive to generate. This limits the number of datasets that can be used for these work. In this sense, the data is highly supervised.”*
>
> We agree that the task at hand requires very detailed and expensive annotations. However, the fact that the OASIS discriminator is also by design a semantic segmentation model could help to reduce the amount of ground truth label maps needed during inference, and potentially during training. Our discriminator can produce label maps from unlabeled images and feed these label maps to the generator to produce new images.  This means OASIS could handle data with missing annotations and can be used in a semi-supervised setting (where only part of training data is annotated) This can significantly reduce the spent annotation effort and extend the number of potential datasets.
>
> To get a better picture of the segmentation performance of the discriminator, we evaluated it on ADE20K and measured an mIoU of 40.0. For comparison, the state-of-the-art in semantic segmentation for ADE20K is an mIoU of 46.91 [1]. This capability of OASIS allows us to take an image without annotations, infer a label map via the discriminator itself and generate new images from the inferred label map using the generator. We visually demonstrate this application in Appendix B4 and Figure K. We consider these experiments evidence that OASIS can be employed in a semi-supervised setting.
>
> $ $
>
> **R4-2:** *“I would love to the results on a more challenging dataset like LVIS and/or Open Images. For Open Images, it would be interesting to see if the performance scales with 350+ classes. For the former, it would be interesting to see how well it does on objects that appear very rarely.”*
>
> Thank you for the suggestion. We would be also interested to see how OASIS performance scales with many classes. When it comes to rare, imbalanced class categories, we already have evidence that OASIS performs better than SPADE+, which is shown in Appendix A3: To measure the performance on rare classes, we first divide the classes into several splits that we sort by rarity. For example, for COCO-stuff the semantic classes in the rarest split together only constitute 1.4% of the total area in the images (e.g. “spoon”), whereas the categories in the least rare split occupy 69.3% of the total image area in the dataset (e.g. “sky”). OASIS outperforms SPADE+ on all splits, but the margin on the rarest split is larger. This means OASIS deals better with imbalanced categories than SPADE+. For example, the mIoU on the least rare split is 51.1 vs 59.0 (SPADE+ vs OASIS), while it is 22.1 vs 40.6 (SPADE+ vs OASIS) in the split that contains the rarest classes. We show in Appendix A.3 and Table B that this improvement in performance is due to our class-balancing in the N+1 loss. The label balancing weighs the loss for each pixel using the inverse label frequency of the corresponding semantic label at this position. In doing so, rarer classes have a higher weight. Consequently, when class-balancing is activated, the mIoU of the rarest labels in ADE20K improves from 31.2 to 39.7.  Note that this per-pixel class-balancing is only possible for OASIS, since only OASIS computes an adversarial segmentation loss. For this reason, we expect OASIS to also perform notably better for rare objects in the long-tailed LVIS dataset.
>
> As the time for experiments on Open Images and LVIS will take comparable or even longer time to the ones on COCO-Stuff (~4 weeks) we are unable to produce them during the rebuttal period. We consider these experiments an interesting future work and thank the reviewer for the suggestion.

---

### Official Review · AnonReviewer3 · 2020-10-28
**Good Improvement over Baselines, but Technical Novelty is Limited**

**Rating:** 6
**Confidence:** 3

**Review:**

In this paper, the authors approach the problem of conditional image generation via generative adversarial networks. To this end, they propose an approach that utilizes only semantic segmentation annotations and adversarial loss. No perceptual loss is required. Their discriminator leverages semantic labels to improve the image generations. They evaluate their approach on a variety of datasets including ADE20K, COCO, and CityScapes. They demonstrate substantial quantitative and qualitative performance over baselines and perform an ablation analysis.

Pros:
1. The problem of mask conditioned image generation has immediate applications in computational photography and computer graphics.

2. The paper is well written and easy to understand.

3. Performance is impressive and experimental evaluation is thorough. The authors perform an ablation analysis.

4. The proposed techniques (Segment based discriminator, 3d noise vector) are useful insights into tuning GANs to perform well.

Cons:
1. Technical novelty is limited. The contributions ( segment discriminator and 3D noise) are useful engineering increments but not particularly large insights into the workings of image generation.

2. While the need for VGG based perceptual loss may not be needed here, it seems like the perceptual loss has just been shifted to the spatial realm. There is still a dependence on strong semantic constraints. Instead of using image class labels to train features, the per pixel class labels are used.

In summary I believe the approach shows promise and important tweaks for image generation networks. Performance is good. However, the technical novelty is limited.

---

> ### Author Response · Authors · 2020-11-17
> **Response to Reviewer 3**
>
> Thank you for the positive feedback! We would like to take the opportunity to share our perspective on the two cons you have raised:
>
> $ $
>
> **R3-1:** *“Technical novelty is limited. The contributions (segment discriminator and 3D noise) are useful engineering increments but not particularly large insights into the workings of image generation.”*
>
> Current GAN models for semantic image synthesis rely on external networks (VGG) for training stability and synthesis quality. While these methods achieve good results, the external supervision imposes model bias and dataset bias. Our contribution is to remove the need for such external supervision. By removing the bias, we improve synthesis quality and diversity.
>
> Thus, through our technical contributions, GAN models for semantic image synthesis can now behave like other cGANs: The model is sensitive to noise and no external supervision is needed.
>
> Lastly, our technical contributions enable new features such as (1) local editing of generated images, (2) segmenting and re-synthesizing real images (see R4-1, App. B4 and Figure K) as well as (3) local class-balancing, which helps training on imbalanced datasets.
>
> $ $
>
> **R3-2:** *“While the need for VGG based perceptual loss may not be needed here, it seems like the perceptual loss has just been shifted to the spatial realm. There is still a dependence on strong semantic constraints. Instead of using image class labels to train features, the per pixel class labels are used.”*
>
> Thanks a lot for your perspective! We would like to provide an argument for why our method reduces supervision, and why the perceptual loss and the adversarial segmentation loss are not equivalent:
>
> By design semantic image synthesis is a task where pairs of label maps and real images are given. Note that all previous works also provide the image and the label map to the discriminator. However, in our work, we make more effective use of the label maps, through the proposed adversarial segmentation loss. This enables the generation of high-quality images without the need for extra supervision from an external perceptual network (VGG). Since the use of VGG was crucial for previous works, but not for OASIS, we reduce the amount of supervision needed.
>
> Further, the use of the perceptual loss and adversarial segmentation loss is not equivalent: the perceptual loss matches features based on what VGG learned from ImageNet, independent of the ground truth semantic labels. In contrast, the adversarial segmentation loss explicitly takes the ground truth semantic labels into account. This also has the immediate benefit of better alignment with the ground truth label maps, as shown by the improvement in mIoU.

---

### Official Review · AnonReviewer1 · 2020-10-29
**Well executed paper, some clarifications are missing.**

**Rating:** 7
**Confidence:** 3

**Review:**

Post-rebuttal:

I acknowledge reading the rebuttal as well as other reviewers comments. I'm satisfied with the rebuttal, I think that the authors have addressed many of my initial comments and I'm happy to increase the score of the paper to 7. If the paper gets accepted to the conference I would encourage the authors to include and expand a discussion about method limitations in the main body of the paper.
----------------------------------------------------------------
Pre-rebuttal:

The paper deals with semantic image synthesis and proposes a model that can generate images from segmentation masks. The main methodological contribution of the paper is a segmentation-based discriminator. By using such discriminator, the model is able to learn high quality image syntheses. The proposed pipeline is evaluated on multiple datasets (ADE20K, Cityscapes and COCO-stuff). The qualitative and quantitative results highlight good image quality of the proposed pipeline.


Pros:

Solid model evaluation including large number of ablations.
Simple and intuitive pipeline design that is shown to produce high quality image syntheses.
Well written paper and well structured appendix.


Cons:

Although the synthesized image quality is improved it seems to come at the expense of image diversity.
The terms used in the papers are sometimes confusing and could be improved.
Although the paper uses standard evaluation metrics for image generation (such as FID), their use for conditional image synthesis is is not fully justified.


Detailed review and comments:

Abstract:
-> "semantic image synthesis GAN" - I'm not convinced with this wording used in the paper. It mixes task (semantic image synthesis) with a type of generative model (GAN). This introduces some confusions in the text, e.g. "semantic image synthesis GAN models still greatly suffer from poor image quality when trained with only adversarial supervision" that might be a bit unclear since GANs are always trained with only adversarial supervision. I would recommend to  simply use "semantic image synthesis".

Introduction:
-> Conditional GANs references: Goodfellow et al 2014 is a good reference for unconditional GAN, not conditional GAN setup. Currently, this reference is linked with other references to conditional reconstruction.

-> Since this is an application oriented paper I think that the introduction would benefit from motivating a bit the task that is under study in this paper. e.g. why semantic image synthesis is of interest to ICLR community?

-> "Although the perceptual loss substantially improves the accuracy of previous methods, it comes with the computational overhead introduced by utilizing an extra network for training. Moreover, it usually dominates over the adversarial loss during training, introducing a bias towards ImageNet, which can have a negative impact on the diversity and quality of generated images, as we show in our experiments." - I'm not convinced if the diversity reduction when using the perceptual loss is due to the bias towards ImageNet. The reduction of diversity is rather an effect of perceptual loss overall. Could the authors comment on this?

Prior work references:
-> This reference (https://arxiv.org/pdf/2004.03590.pdf) seems relevant. Could the authors comment on this reference in the paper? Adding comparison to this work would make the paper even stronger.

Methodology:
-> I find the wording of "3D noise sampling" a bit unclear. Is it just a resized version of 1D tensor? Is there any importance on how the 3D noise sampling is structured? Could the authors clarify the idea of 3D noise sampling in the main body of the paper?

Experiments:
-> Could the authors comment why FID is a good metric for semantic image synthesis? Justifying the choice of metrics would make the evaluation section stronger.  From the discussion of the results, (diversity quality tradeoffs) it seems that the paper would benefit from reporting precision-recall like metrics for GANs (e.g. https://arxiv.org/abs/1807.09499, https://arxiv.org/abs/1905.10887, https://arxiv.org/abs/1904.06991). Adding additional metrics would further improve the paper.

-> Table 2. It is unclear to me what is the meaning of red color numbers. Could the authors add clarifications about it to the caption?

-> Table 2. It is bit surprising to see that the model trained only with adversarial loss has relatively low diversity. I would expect that the lack of perceptual loss would lead also to increased diversity in synthesized images. Could the authors comment on what is the limiting factor to increase sample diversity? Could the diversity be affected by the proportion of mixing (real vs. fake)  in LabelMix?

-> Would it be possible to add qualitative results for SPADE+? This model seems to have high diversity scores and it would be interesting to see model samples.

-> The paper lacks discussion on the limitations of the method. Could the authors comment on the potential limitations of the suggested approach?

---

> ### Author Response · Authors · 2020-11-17
> **Response to Reviewer 1**
>
> **R1-10:** *“This reference (https://arxiv.org/pdf/2004.03590.pdf ) seems relevant. Could the authors comment on this reference in the paper? Adding comparison to this work would make the paper even stronger.”*
>
> Thanks for pointing towards that reference. The referenced paper proposes an interesting approach to semantic synthesis, based on the relatively novel principle of implicit maximum likelihood estimation. We now comment on this reference in the revision in the related work section on page 3. However, direct quantitative comparison to OASIS and preceding work such as SPADE and CC-FPSE is difficult for several reasons: (1) The paper uses a different evaluation protocol. Instead of employing FID and mIoU for evaluation, it uses LPIPS and crowd-sourced human judgment. (2) This work does not also evaluate on the standard label-to-image benchmarks, such as ADE20K, Cityscapes and COCO, but instead uses the GTA5 dataset and the BDD100K dataset. (3) The proposed model is not quantitatively compared to state-of-the-art GAN models for semantic image synthesis.
>
> The closest possible comparison is to visually compare their BDD100K samples to the Cityscapes samples of OASIS, since both datasets consist of real urban street scenes. From visually inspecting Fig. 14 in the given reference, it is evident that the synthesized street scenes have blurry textures and lack detail. In contrast, OASIS synthesizes less blurry images with more realistic colors and a lot more detail, such as windows on house facades and driving lanes on the streets (see Appendix Fig. D).
>
> $  $
>
> **R1-11:** *“I find the wording of "3D noise sampling" a bit unclear. Is it just a resized version of 1D tensor? Is there any importance on how the 3D noise sampling is structured? Could the authors clarify the idea of 3D noise sampling in the main body of the paper?”*
>
> During training, the 3D noise tensor is sampled globally, i.e. per-channel, where we sample each channel value and copy it along the height (H) and width (W) dimension of the tensor.  In other words, globally sampled 3D noise can be viewed as a spatially replicated version of a 1D tensor. We have also investigated other ways of structuring the 3D noise tensor during training in Appendix A7. Based on our analysis we chose global sampling for training, due to its simplicity and competitive performance with other considered strategies. We added a clarification about the 3D noise sampling in section 3.3 on page 6.
>
> Note that during inference, the 3D noise can be sampled in various ways: (1) globally as during the training, i.e. sampling per-channel and replicating each channel value spatially to the whole tensor; (2) locally, sampling different channel values area-wise. The option (1) entirely changes the image. The option (2) allows us to only change parts of the image. For example, based on the label map, we can extract the area of sky. Through the option (2), different noise realizations in the area corresponding to the sky segment lead to different skies in the generated images without changing the rest of the scene (see Fig. F in the Appendix or Fig.2).
>
>  $ $
>
> **R1-12:** *“Table 2. It is unclear to me what is the meaning of red color numbers. Could the authors add clarifications about it to the caption?”*
>
> Table 2 reports the results of the ablation on the use of VGG and 3D noise, showing their impacts on the diversity and quality performance of SPADE+ and OASIS. As we observe a trade-off between diversity and quality, it is important to jointly assess a model from both aspects. Therefore, we make not only the best numbers bold, but also mark the worst numbers in red. For instance, SPADE+ with 3D noise without VGG has the best diversity performance (bold), but FID and mIoU are strongly compromised (red). This indicates that the diversity gain is spurious, being a side effect of a poor quality of synthesized images that are naturally more different from each other. OASIS with 3D noise and VGG, has the best mIoU performance (bold), but the diversity is worse (red) than the same version w/o VGG. This indicates that the model sacrifices variety to generate easy-to-segment images. We added a clarification in the caption.
>
>  $ $
>
> **R1-13:** *“Would it be possible to add qualitative results for SPADE+? This model seems to have high diversity scores and it would be interesting to see model samples.”*
>
> Thanks for the suggestion. We added qualitative results for SPADE+ (with VGG) in Appendix Figure E. Please note that despite its improved quality over original SPADE, this model still depends on the VGG loss and has very low diversity (The MS-SSIM for SPADE+ in Table 2 is 0.85, while OASIS has 0.65, which indicates that SPADE+ has lower diversity).

---

> ### Author Response · Authors · 2020-11-17
> **Response to Reviewer 1**
>
> **R1-6:** *“Although the paper uses standard evaluation metrics for image generation (such as FID), their use for conditional image synthesis is not fully justified.” /” Could the authors comment why FID is a good metric for semantic image synthesis? Justifying the choice of metrics would make the evaluation section stronger.”*
>
> FID is arguably the most broadly used metric for evaluating the image synthesis performance of both unconditional (e.g. StyleGAN) and conditional GANs (e.g. SA-GAN, BigGAN, SPADE), as it is sensitive to both quality and diversity of synthesized images. Yet, we agree that FID alone is insufficient for semantic image synthesis, since it is insensitive to the correct alignment of images with their semantic label maps. For this, the mean intersection of union (mIoU) was used by prior work. For fair comparison we follow the evaluation protocol of the preceding semantic image synthesis work and measure both FID and mIoU. In the revision, we now added a justification of the used metric in section 4 on page 7.
>
> As additional metrics, we measure MS-SSIM and LPIPS in Table 2, quantifying the diversity of multi-modal images. Further, in the revision, following your suggestion we added an evaluation on precision and recall as well, see answer R1-7 for details.
>
> $ $
>
> **R1-7:** *“ From the discussion of the results, (diversity quality tradeoffs) it seems that the paper would benefit from reporting precision-recall like metrics for GANs (e.g. https://arxiv.org/abs/1807.09499,  https://arxiv.org/abs/1905.10887,  https://arxiv.org/abs/1904.06991). Adding additional metrics would further improve the paper.”*
>
> Thank you very much for this suggestion. We decided to measure the improved precision and recall metric of [1]. Precision and recall serve as proxies for synthesis quality and diversity. The other two metrics of [2,3] measure the accuracy of a classifier trained on either real or fake data and are therefore rather suited for class-conditional image synthesis.
>
> We present the precision and recall scores in Appendix A9 and Table K. The result is that OASIS outperforms SPADE+ in both metrics for all datasets. For example, on ADE20K we measure a precision and recall of 0.77 and 0.57 for OASIS, whereas SPADE+ reaches 0.71 and 0.52.
>
> [1] Kynkäänniemi, Tuomas, et al. "Improved precision and recall metric for assessing generative models." Advances in Neural Information Processing Systems. 2019.
>
> [2] Shmelkov, Konstantin, Cordelia Schmid, and Karteek Alahari. "How good is my GAN?." Proceedings of the European Conference on Computer Vision (ECCV). 2018.
>
> [3] Ravuri, Suman, and Oriol Vinyals. "Classification accuracy score for conditional generative models." Advances in Neural Information Processing Systems. 2019.
>
>  $ $
>
> **R1-8:** *“semantic image synthesis GAN" mixes task (semantic image synthesis) with a type of generative model (GAN). This introduces some confusions in the text, e.g. "semantic image synthesis GAN models still greatly suffer from poor image quality when trained with only adversarial supervision" that might be a bit unclear since GANs are always trained with only adversarial supervision. I would recommend to simply use "semantic image synthesis."*
>
> We agree that semantic image synthesis is a task and GAN is a type of generative models. We use “semantic image synthesis GAN” to emphasize that our solution for semantic image synthesis targets GAN based models. To avoid potential confusion, here we updated the sentence in the paper as "GAN models for semantic image synthesis”  in the abstract on page 1. We also changed a similar formulation in the introduction on page 2 for clarity. You are right that GANs are trained with adversarial supervision, but often not only using the adversarial loss.  For instance, SPADE and other prior work rely on the extra VGG-based perceptual loss for semantic image synthesis to achieve good synthesis quality.
>
> $ $
>
> **R1-9:** *“The introduction would benefit from motivating a bit the task that is under study in this paper.”*
>
> Semantic image synthesis has several use cases. For example, (1) it can be used to render photorealistic images from semantic layout in professional graphics applications. (2) Since the task gives a lot of fine-grained control to the user, it can be used to create artificial training data that needs to meet specific requirements. An example is the synthesis of rare traffic situations to create additional training data for a segmentation network used for autonomous driving. We revised the introduction with a motivation for the semantic image synthesis task on page 1 and 2.

---

> ### Author Response · Authors · 2020-11-17
> **Response to Reviewer 1**
>
> Thanks a lot for your detailed comments and valuable feedback.  We appreciate your review and have made changes in the paper to better address your concerns. We next address the individual points:
>
> $ $
>
> **R1-1:** *“Although the synthesized image quality is improved it seems to come at the expense of image diversity.”*
>
> We would like to clarify that our method increases the diversity as well as improves the image quality. We measure diversity with the MS-SSIM and LPIPS metrics and show the results in Table 2: SPADE+ has an MS-SSIM of 0.85, while OASIS has 0.65. This means that images generated by OASIS are more diverse. Please note that lower MS-SSIM means higher diversity, as indicated by the up/down arrows in Table 2. We added this clarification about the diversity metrics in the updated paper in section 4.1 on page 8.
>
> Furthermore, we observe a trade-off between image synthesis quality (FID and mIoU) and diversity (MS-SSIM and LPIPS) for SPADE+. We visualize this trade-off in Table 2 by highlighting the best results in bold and the worst in red. We observe that only OASIS w/o VGG loss achieves high quality and diversity at the same time. Compared to SPADE+ with image encoder, OASIS achieves better quality (28.3 vs 33.4 FID, 48.8 vs 40.2 mIoU) and better diversity (0.65 vs 0.85 MS-SSIM, 0.35 vs 0.16 LPIPS).
>
> $ $
>
> **R1-2:** *“It is bit surprising to see that the model trained only with adversarial loss has relatively low diversity. I would expect that the lack of perceptual loss would lead also to increased diversity in synthesized images. Could the authors comment on what is the limiting factor to increase sample diversity? Could the diversity be affected by the proportion of mixing (real vs. fake) in LabelMix?”*
>
> We would kindly point out that the model trained with only the adversarial loss increases diversity, not decreases it. Hence, the expectation that the lack of perceptual loss also increases diversity is in line with our results. Please see our answers to the previous question R1-1 for details.
>
> LabelMix is a tool to improve synthesis quality through consistency regularization. We use a 50-50 real/fake mixing proportion, which is intended to ensure good synthesis quality at the semantic boundaries. Thus, Table A shows an average improvement of 2.2 FID and 2.25 mIoU through LabelMix. However, LabelMix is not designed to increase diversity and slightly shifts MS-SSIM up, by 0.015 points on average in Table A. Therefore, we do not think that changing the mixing proportion will increase diversity, but it may weaken the quality gain obtained by LabelMix.
>
>  $ $
>
> **R1-3:** *“I'm not convinced if the diversity reduction when using the perceptual loss is due to the bias towards ImageNet. The reduction of diversity is rather an effect of perceptual loss overall. Could the authors comment on this?”*
>
> Yes, the reduction of diversity is an effect of the perceptual loss. Unlike the discriminator (which adapts the discriminative features along with the learning progress of the generator), the VGG features are static and do not change in response to the generator. As a result, the generator can easily get stuck at some dominant modes imposed by the VGG features space. This feature space is constrained by both, the design of the VGG architecture and the data used for VGG training. Since VGG is trained on ImageNet, its features are biased towards ImageNet. However, if VGG was trained on a different dataset, the reduction in diversity might still occur. To be more precise, we re-wrote the sentence in the introduction on page 2: "Although the perceptual loss substantially improves the accuracy of previous methods, it comes with the computational overhead introduced by utilizing an extra network for training. Moreover, it usually dominates over the adversarial loss during training, which can have a negative impact on the diversity and quality of generated images, as we show in our experiments."
>
>  $ $
>
> **R1-4:** *“Conditional GANs references: Goodfellow et al 2014 is a good reference for unconditional GAN, not conditional GAN setup”*
>
> We put the Goodfellow '14 reference together with the conditional GAN references to give credit to the original GAN paper. However, since it may cause confusion we removed it in the updated version.
>
> $ $
>
> **R1-5:** *“The paper lacks discussion on the limitations of the method. Could the authors comment on the potential limitations of the suggested approach?”*
>
> We observe that OASIS has more diversity in colors and textures than SPADE and the related work CC-FPSE. This can sometimes lead to unnatural outlier colors and textures. Examples of such failure cases are presented in Figure G.

---

### Decision · Program_Chairs · 2021-01-07
**Final Decision**

**Decision:**

Accept (Poster)

**Comment:**

The paper received 3 reviews with positive ratings: 7,6,7. The reviewers appreciated overall quality of the manuscript, thoroughness of the evaluation, and practical importance of this work (mentioning though that the technical novelty is still not high). They also acknowledged impressive empirical performance. The authors provided detailed responses to each of the reviews separately, which seemed to have resolved the remaining concerns.
As a result, the final recommendation is to accept this work for presentation at ICLR as a poster.